# Modeling the START transition in the budding yeast cell cycle

**Janani Ravi** [1] *, **Kewalin Samart** [1,2], **Jason Zwolak** [3]

**1** Department of Biomedical Informatics, University of Colorado Anschutz Medical Campus, Aurora, Colorado, United States of America, **2** Computational Bioscience program, University of Colorado Anschutz Medical Campus, Aurora, Colorado, United States of America, **3** InSilica Labs, Asheville, North Carolina, United States of America

* janani.ravi@cuanschutz.edu

## Abstract

Budding yeast, *Saccharomyces cerevisiae*, is widely used as a model organism to study the genetics underlying eukaryotic cellular processes and growth critical to cancer development, such as cell division and cell cycle progression. The budding yeast cell cycle is also one of the best-studied dynamical systems owing to its thoroughly resolved genetics. However, the dynamics underlying the crucial cell cycle decision point called the START transition, at which the cell commits to a new round of DNA replication and cell division, are under-studied. The START machinery involves a central cyclin-dependent kinase; cyclins responsible for starting the transition, bud formation, and initiating DNA synthesis; and their transcriptional regulators. However, evidence has shown that the mechanism is more complicated than a simple irreversible transition switch. Activating a key transcription regulator SBF requires the phosphorylation of its inhibitor, Whi5, or an SBF/MBF monomeric component, Swi6, but not necessarily both. Also, the timing and mechanism of the inhibitor Whi5's nuclear export, while important, are not critical for the timing and execution of START. Therefore, there is a need for a consolidated model for the budding yeast START transition, reconciling regulatory and spatial dynamics. We built a detailed mathematical model (START-BYCC) for the START transition in the budding yeast cell cycle based on established molecular interactions and experimental phenotypes. START-BYCC recapitulates the underlying dynamics and correctly emulates key phenotypic traits of ~150 known START mutants, including regulation of size control, localization of inhibitor/transcription factor complexes, and the nutritional effects on size control. Such a detailed mechanistic understanding of the underlying dynamics gets us closer towards deconvoluting the aberrant cellular development in cancer.

## Author summary

Researchers use the budding yeast *Saccharomyces cerevisiae* as a model to understand how complex eukaryotic cells grow and divide, and how important cell cycle-related diseases like cancer progress. The *S. cerevisiae* cell cycle entails carefully controlled growth and

**Data Availability Statement:** All the simulation data and visualizations (for wildtype and 100s of mutants) are available in our interactive online simulator: http://www.sbmlsimulator.com/

simulator/by-start. Our code, differential equation model, and parameters are available via GitHub: github.com/jravilab/start-bycc.

**Funding:** This work was supported in part by the University of Colorado Anschutz start-up funds (awarded to JR; partially supported JR and KS). The funders had no role in study design, data collection and analysis, decision to publish, or preparation of the manuscript.

**Competing interests:** The authors have declared that no competing interests exist.

division determined by the budding yeast genome, as well as factors like the availability of sugars the yeast uses to grow. Scientists have studied these processes for decades and elucidated key details and regulatory mechanisms. Computational models have been built based on this knowledge to analyze and predict the dynamics of the yeast cell cycle and its dependence on environmental factors. Most models to date lack detail at the initial point called the START transition, where cells commit to reproducing by dividing. We designed a new mathematical model called START-BYCC that incorporates the many different regulatory mechanisms by which yeast cells decide whether to enter the START transition. Our model accurately predicts experimental results capturing how ~150 mutant yeast strains (minor variations in genetic background) behave in the START transition. The START-BYCC model can be used to make accurate predictions for the cell cycle and provide insight into the role of this essential transition in diseases like cancer.

## Introduction

### Background and significance

The cell cycle underlies all biological growth, reproduction, development, and repair, and its misregulation results in complex human diseases, such as cancer. Genetic and molecular research over the past several decades has provided a rigorous understanding of the molecular mechanism coordinating the cell cycle [1–4]. The cell cycle growth-and-division process is characterized by alternating DNA replication (S phase) and mitosis (M phase), interspersed by temporal gap phases G1 and G2 to ensure balanced growth and division. Several checkpoints occur throughout the cell cycle progression and control the commitment to DNA replication, proper alignment, and segregation of chromosomes and mitosis.

The molecular machinery of the cell cycle is highly conserved across eukaryotes [5], and much of the regulatory system has been systematically worked out in the budding yeast (*Saccharomyces cerevisiae*). Budding yeast is an especially attractive model organism for study of the cell cycle, largely because it is genetically tractable and can exist as a haploid. In eukaryotes, cyclin-dependent kinases (CDKs) play a key role in initiating crucial cell cycle events by phosphorylating specific protein targets. CDK levels remain constant throughout the cell cycle, but their activity and substrate specificity are governed by their binding partners, cyclins, whose concentrations fluctuate throughout the cell cycle. The cyclin/CDK complexes are tightly regulated by synthesis, degradation, and/or sequestration by the cyclin-dependent kinase inhibitor (CKI) complexes. The budding yeast presents a less complex cell cycle circuitry than mammalian counterparts, with only one CDK (Cdc28) [1] that binds to one of nine cyclins of type Cln or Clb. The most important cyclins involved in the different phases of a budding yeast cell cycle are Cln3 (G1), Cln1,2 (G1/S), Clb5,6 (S), Clb3,4 (early M), and Clb1,2 (late M) (**S1 Table**). The names of the cyclin-binding partners during different phases of the cell cycle are used to refer to the relevant heterodimer.

The yeast cell division process begins with the cell budding and then dividing asymmetrically at the neck to produce a large mother and a small daughter cell, each with a set of sister chromatids. Soon, the mother cell repeats the process; in contrast, the daughter cell has a long G1 phase before producing the first bud and entering the S phase. This characteristic commitment step in the budding yeast cell cycle, including bud initiation, the onset of DNA synthesis, and spindle pole body duplication, is referred to as 'START' [6].

Classic studies on 'size control' in budding yeast have shown that small newborn daughter cells in G1 have longer lag periods than larger newborn cells before the START transition (first

appearance of bud), even if they are genetically identical and placed under identical physiological conditions [7]. This delay allows the small cells to grow large enough to attain a minimum size threshold before their START transition and commitment to the S phase. Notably, this size threshold depends on the nutrient medium, with the threshold increasing in proportion to the richness of the medium [8–11]. Cells passing through the START transition also abruptly lose their response to pheromones (mating factors), which are inhibitors of the cell cycle [8–10] and to which cells are sensitive in the G1 phase. Contingent on favorable external (nutrients, pheromones) and internal (DNA damage) cues, START is the point of commitment towards coordinated DNA replication and cell division [11,12].

## Key dynamical processes modeled

The most important aspects underlying the molecular mechanisms for the budding yeast cell cycle have been studied in great detail [13–15]. We previously built mathematical models of the budding yeast cell cycle [16,17] based on these extensive molecular studies. Here, we build upon a detailed mechanistic cell cycle model by supplementing the most relevant findings in START.

*START transition*: In early G1, only Cln3 is available. The dynamics of Cln3 in this context have been a subject of investigation and debate in the field for decades: from its initial characterization (as WHI1), Cln3 was reported to be linked to cell size [18], and some current hypotheses continue to support that Cln3 levels increase through cell size-dependent mechanisms [19], while others have shown intrinsic metabolism-dependent bursts of Cln3 production [20]. When the cell attains a critical size, Cln3 activates two transcription factors: SBF, a heterodimer of Swi4 and Swi6 [21], and MBF, a heterodimer of Mbp1 and Swi6 [22]. In the absence of Cln3, Bck2 plays a role in the activation of SBF and MBF in response to cell size [21,23]. SBF and MBF drive the irreversible START transition (except under starvation [24]) by activating the transcription of G1 cyclins Cln1,2, and S phase cyclins Clb5,6 [21,23,25]. SBF and MBF have a large functional overlap [26]. In the absence of SBF, MBF can activate Cln1,2. Likewise, SBF can activate Clb5,6 in the absence of MBF.

Clb5,6 are not active initially in late G1 due to the presence of their stoichiometric CDK inhibitor, CKI (Sic1 and Cdc6) [27]. Active Cln1,2 phosphorylate Sic1 (and Cdc6), signaling it for rapid degradation [28], resulting in Clb5,6 activation soon after Cln1,2 activation. At START transition, as SBF and MBF are activated, Cln1,2 accumulate first, leading to bud emergence, followed closely by Clb5,6 activation, leading to the initiation of DNA synthesis (**Fig 1A**). In wildtype cells, these two events occur almost simultaneously–a characteristic of START transition [6].

*G2/M transition*: Clb5,6 activated at START inhibits Cdh1, a protein that degrades mitotic cyclins Clb1,2 [29]. Thus, Clb1,2 accumulate when their synthesis is turned on later in the G2 phase, followed by entry into mitosis (**Fig 1A**). Thus, the START transition also facilitates the progression of the cell cycle to the G2/M phase [30]. At the end of the S phase, SBF is turned off by Clb1,2 [31,32], and MBF by Clb1,2 and Nrm1 [33].

*FINISH transition*: At the end of mitosis, when all sister chromosome pairs are attached to opposite poles of the spindle, Cdc20 is activated [34]. Cdc20, along with the anaphase-promoting complex (APC), causes the dissolution of the cohesin that holds the sister chromatids together, allowing them to move apart to the opposite poles of the spindle. The dissolution of chromatids triggers the activation of a phosphatase, Cdc14, known for three key roles: (i) activation of CKI synthesis [35], (ii) dephosphorylation and stabilization of CKI [36], and (iii) activation of Cdh1 with the help of Cdc20, leading to the degradation of Clb1,2 [37–39]. Degradation of mitotic cyclins enables the cell to exit from mitosis, resetting the cell for the

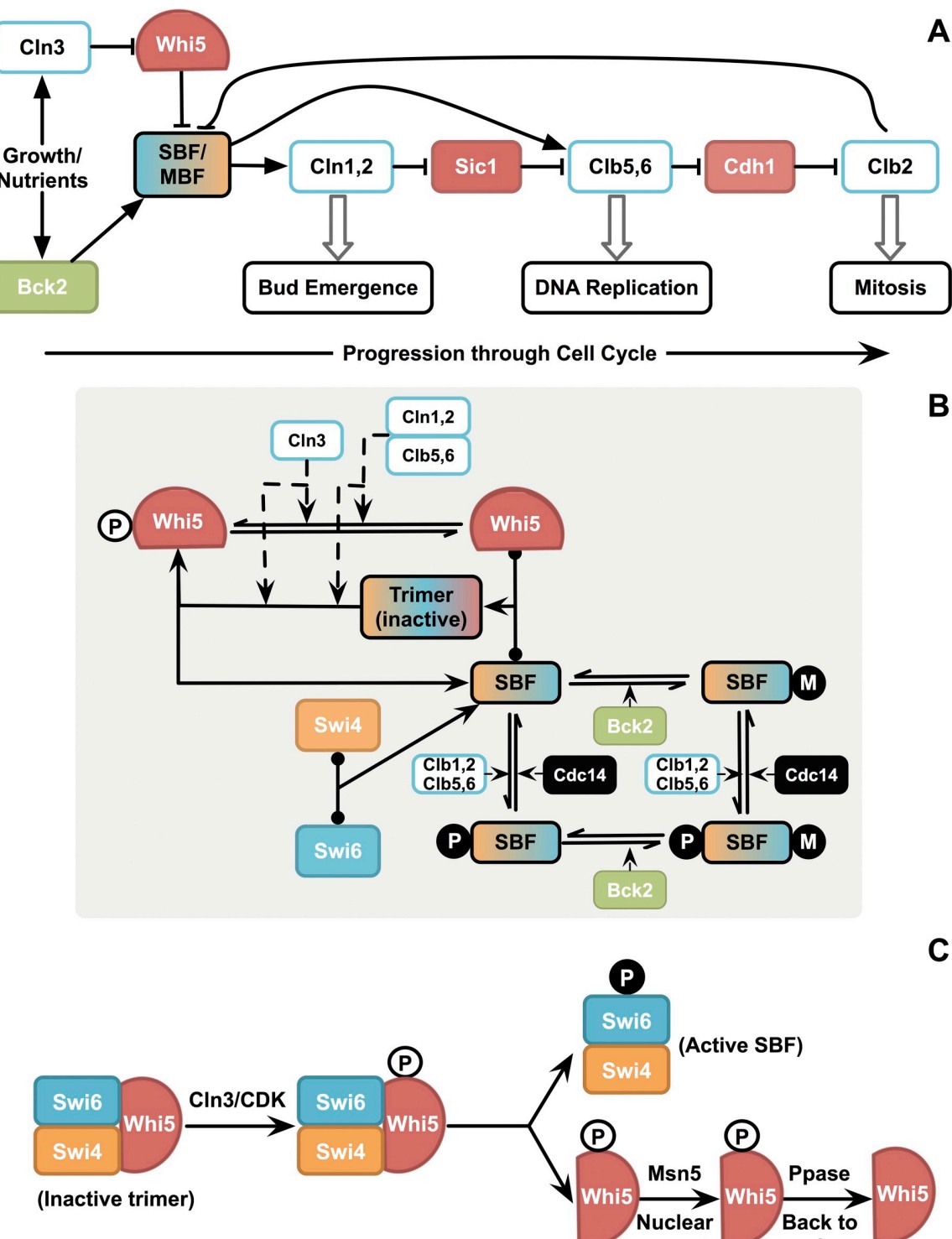

**Fig 1. Simplified mechanism of cell cycle progression and START transition.** (A) Progression through the cell cycle in budding yeast. Cln3 and Bck2 are activators of START (turn on SBF, MBF needed for Cln1,2; Clb5,6). Cln1,2 phosphorylates and inhibits Sic1, a stoichiometric inhibitor of Clb5,6, thus allowing DNA replication to occur. S-phase cyclins, Clb5,6, inhibit Cdh1, an antagonist of the mitotic cyclins, Clb1,2, thus allowing progression through the mitotic events, and finally exit from mitosis leading back to G1. (B) The earlier hypothesis that Whi5 phosphorylation is crucial for SBF activation. This model for START includes (i) activation of SBF by Clns (Cln3, Cln1,2, Clb5,6) by inactivation of Whi5 (by phosphorylating free and SBF-bound Whi5) in late G1, (ii) activation of SBF by Bck2 to an alternate form independent of Whi5 and CDK, and (iii) inactivation of SBF by Clb1,2 in late S phase. (C) Earlier hypothesis on association, dissociation, and translocation events underlying SBF activation. Inactive SBF-Whi5 trimer gets phosphorylated by Cln/CDK on both Swi6 and Whi5, followed by the trimer dissociation to give active SBF (with Swi6 phosphorylated) and phosphorylated Whi5 that can get exported to the cytoplasm (by transport protein, Msn5).

next cycle [40]. The details of the metaphase-to-anaphase transition involving other critical proteins such as Esp1, Pds1, MET, and other proteins are beyond the scope of this model, but are incorporated in detail in Hancioglu and Tyson 2012 [41]. In summary, at FINISH, the cell exits mitosis and returns to G1 if DNA is fully replicated and undamaged, with chromosomes aligned perfectly at the metaphase plate, ready to be separated into daughter cells [42]. The mechanistic details underlying the FINISH transition have been studied and addressed by various mathematical models [41,41,43–45], including the latest developed model combining the deletion of other genetic elements of the FINISH transition with Clb2 overexpression to predict the presence or absence of Cdc14 endocycles in mutant strains [46].

*START vs. R-point*: Activation of SBF by Cln3 is central to START, and prior studies suggest that a double-repression mechanism underlies this activation (**Fig 1A–1B**). SBF is bound and inactivated by the inhibitor, Whi5 [47,48]. Cln3 to (Cln3/Cdc28) phosphorylates Whi5, which causes its dissociation from SBF, rendering SBF active (**Fig 1B**). Bck2, another known activator of START, acts by a mechanism independent of CDK and Whi5 [49]. In the late S phase, SBF is phosphorylated and turned off by Clb2 [31,32] and Clb6 [50] (**Fig 1B**). It is noteworthy that the Cln3-Whi5-SBF (cyclin-inhibitor–transcription factor) circuit that operates at the START transition in budding yeast is thus reminiscent of the CycD-pRb-E2F system operating at the restriction point (R-point) in mammals, albeit with different dynamics: [Cln3 –| Whi5 –| SBF → Cln1,2 → START] vs.[CycD–| pRb–| E2F → CycE → R-point] [51]. The inhibitor pRB is a known tumor suppressor, and genes resulting in cyclin activation are known oncogenes [52–55], with pRB phosphorylation playing a key role in G1/S transition [56].

*Whi5 phosphorylation*: Due to the parallels between the mammalian R-point and yeast START systems, the prevailing notion was that the timely activation of START solely depends on the phosphorylation of Whi5 and resulting SBF activation (**Fig 1C**). As cells progress through START, Whi5 gets progressively phosphorylated and inactivated [57], with a nonlinear accumulation of Whi5 [58]. Consequently, it becomes cytoplasmic in late G1, leaving SBF in the active state for Cln1,2 transcription. Whi5 moves back to the nucleus to inhibit SBF only at mitotic exit. Phosphorylation was also thought to play a role in the nuclear export of Swi6 in the late S phase concurrent with SBF inactivation [50,59]. These observations together made up the simple story of the activation/inactivation of SBF and the relocalization of the inhibitor (**Fig 1C**). Accordingly, if the phosphorylation of Whi5 was key to its export and SBF activation, then mutating all the phosphorylation sites (to alanine) should retain Whi5 in the nucleus, delaying SBF activation, and hence resulting in larger cells. Likewise, mutation of the phosphorylation sites on Swi6 should cause Swi6 retention in the nucleus, thereby advancing START and resulting in smaller cells.

The study by Wagner et al. [60], however, challenged both of these expectations. *WHI5-12A*, a non-phosphorylable mutant with all known CDK (and non-CDK sites) phosphorylation sites in Whi5 mutated to alanine, showed no difference in size compared to wildtype [60]. *SWI6-SA4*, a mutant with non-phosphorylable Swi6, was also observed to have a wildtype size [59,61]. Only the double non-phosphorylable mutant *WHI5-12A SWI6-SA4* showed a 40% increase in size [60]. Together, these experiments indicate that either Whi5 or Swi6 needs to be phosphorylated (if not both) for timely activation of SBF (so that there is no difference in size). These observations present a significant departure in the START transition process from that of the mammalian restriction point. While Whi5 phosphorylation is not necessary for the activation of SBF in budding yeast, pRb phosphorylation has been shown to be absolutely crucial for the activation of E2F in mammalian cells [47,62]. Another challenge to the canonical model for Whi5 phosphorylation by Cln3 notes that Whi5 as a substrate for Cln3-Cdk1 has lower affinity than RNA polymerase II subunit Rpb1 –in this model, phosphorylation by Cln3-Cdk1 of Rpb1

promote SBF-dependent transcription instead of Whi5 being a primary factor [63]. This suggests that Whi5 and Cln3 may act through independent mechanisms to modulate SBF activity.

Whi5 is synthesized fairly consistently, at the highest rate during S/G2/M phases and independently of cell size [64,65]. Its concentration and nuclear localization are critical to its role as a cell cycle inhibitor [64]. Differing Whi5 concentrations have been shown to influence the duration of pre-START G1, and mother cells expressing more Whi5 produce larger daughter cells in a manner independent of Cln3 levels, supporting a role as a sizer [64]. Whi5 is also sequestered so daughter cell Whi5 concentrations are higher than the mother cell [64]. An interplay of activator Cln3 (expression of which may be linked to cell size, and titration of which is against a finite number of genomic binding sites) and inhibitor Whi5 (produced steadily, but which is diluted as cell size increases) is thought to link cell cycle progress with cell size [64,66].

*Modeling the yeast cell cycle*: Mathematical modeling has proven to be an invaluable tool for understanding the workings of cellular regulatory systems such as the cell cycle [42,67,68]. The budding yeast cell cycle has been studied and is understood in great detail using the deterministic mathematical models published by Chen et al., [16,17] (using ordinary differential equations). The 2004 model [17] (henceforth referred to as the BYCC model), in particular, incorporates several aspects of the cell cycle mechanism from START transition through mitotic exit and can explain the phenotypes of ~120 cell cycle mutants. The model, however, contains a very simplistic depiction of START: SBF/MBF activation by Cln3 and Bck2, and inactivation by Clb2 in a condensed phenomenological abstraction (using an ultra-sensitive switch [69]). It does not include several of the more recently discovered details pertaining to START discussed above.

Subsequently, we extended the BYCC model to incorporate the proposed role of Whi5 and its impact as a sizer based on its dilution as cell volume increases, and the relevant functional localization between nucleus and cytoplasm, enabling deeper insights into the critical cell size required for the G1/S transition and budding [66]. A few recent models from our group and others have included more mechanistic details for cell size control [70,71] and viability [66,72–74], including the separation of Clb5,6 into Clb5, Clb6 and emphasizing the positive feedback loops in the START transition [72]. Genome-scale models have applied reaction-contingency language [75], while we also included aspects of START and FINISH modules [42,46]. Concurrently, we also built alternate and abstracted START models with a standard component model (SCM) and Boolean approaches leveraging continuous, discrete, and stochastic methods [76,77]. Other studies have focused on the cell fate decision at G1 involving the mating factor Far1 [78] and the feedforward regulation-mediated loop [79]. Additionally, a recent study discovered that the START transition could be reversed under starvation by interrupting the positive feedback loop that activates the G1/S transition, resulting in re-importing Whi5 to the nucleus to inhibit SBF [24]. In summary, while we and others have developed and derived insights from several kinds of models representing various stages of the cell cycle, we still lack a detailed understanding of the molecular mechanisms underlying START dynamics.

In light of all the findings of the role of Whi5 phosphorylation and their discrepancies with the previously held consensus mechanisms, there is a need to reconcile a coherent model for the START transition in budding yeast that recapitulates the START dynamics while staying consistent with the observed genetics and mutants. Such a model would offer broad insights into the study of cell cycle dynamics and molecular mechanisms, as well as address specific questions about the dynamics underlying the BYCC START transition. In this paper, we have developed a comprehensive nonlinear ordinary differential equation mathematical model for START, with the following features: i) a detailed mechanism for the activation and inactivation

of SBF, along with the inhibitor Whi5, that is compliant with the experimentally determined phenotypes [47,80], ii) a mechanism for activation and inactivation of MBF [21,33,48], iii) highlighting the role of Bck2 in the START transition, iv) critical aspects of the localization of the monomers in the transcription-factor/inhibitor complex (Whi5, Swi6, and Swi4) [57,59,81], and, v) size control operating at the START transition [80,82], and we have also taken into account the specific contribution of Cln3 in setting the cell size threshold [83].

Our new model for START has been integrated with the full cell cycle model [17] (BYCC), resulting in START-BYCC. Our current model (START-BYCC) explains wildtype cell-cycle dynamics, timely localization of monomers, and size control; it elucidates prior experimentally observed phenotypes that lacked a mechanistic explanation, and predicts phenotypes of over ~120 START mutants, a few of which have since been experimentally corroborated [46].

## Results and discussion

We used the detailed BYCC mathematical model [17] as a starting point and built a detailed model for START transition upon it by reconciling several recent studies (**S2 Table** and **Figs 2** and **3**).

The main focus of START-BYCC is the description of the activation and inactivation of G1/S transcription factors, SBF and MBF. At its core, the model contains the SBF and MBF monomers Swi4, Swi6, and Mbp1, and the inhibitor protein, Whi5, levels of which (as with most other model variables) can be adjusted in the online simulator (http://www.sbmlsimulator.com/simulator/by-start). Additionally, we consider two distinct promoters for the two sets of genes involved in budding and DNA synthesis that are turned on by SBF and MBF, respectively. We take into account all pools of monomers and protein complexes that are either bound or unbound to the promoter. We also incorporate all the pertinent phosphorylation states for each of these components based on whether they are Cln/CDK or Clb/CDK targets. Further, we include any available information on the cellular localization of these molecules during different phases of the cell cycle.

Our current model of the cell cycle with a detailed START transition (START-BYCC; github.com/jravilab/start-bycc) is very complex, highlighting the key molecular mechanisms. The START transition subsystem of the model (SBF/MBF regulation) includes 51 species and 56 parameters (**S4 Table**), compared to 1 species and 8 parameters in BYCC, allowing us to simulate the START dynamics and yeast genetic system (with knockouts and overexpression) in such detail for the first time. To systematically detail the critical changes we have made, we describe and discuss each new aspect of our current model of START transition in the subsequent sections:

a. a detailed mechanism for the activation and inactivation of SBF (involving Whi5), MBF, and the role of Bck2 in wildtype cells,

b. the timing and localization of different monomers corresponding to activation and inactivation of SBF (Swi4, Swi6, Whi5),

c. the role of phosphorylation in START transition, and

d. the mechanism for cellular size control involving concentration of Whi5, Ydj1, and Ssa1.

We present the cell cycle dynamics and phenotypes of all relevant cell cycle mutants (200+, including >100 START mutants and several critical FINISH mutants from BYCC) as timecourse simulations (See *Methods* and **S4** and **S5 Tables** for details). We successfully fit 95% of the mutant phenotypes (**S5 Table**) and describe in some detail key underlying mechanistic aspects of the model and challenges (See *sections below*). We also make a few predictions based on our current detailed model for the budding yeast START transition (**Table 1**).

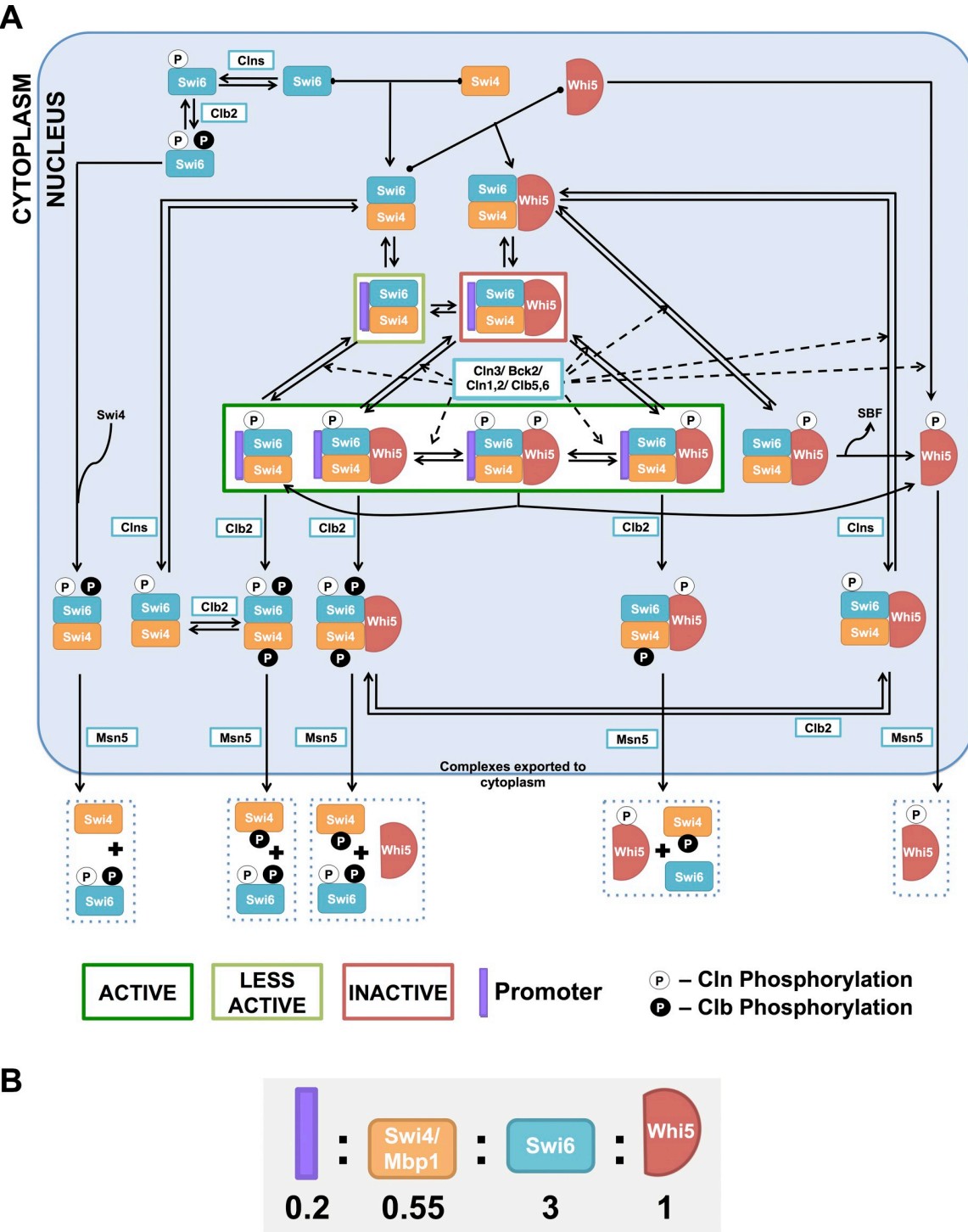

**Fig 2. SBF regulation.** (A) The core model of SBF activation and inactivation by Clns and Clbs. Presented in the figure are the most important interactions considered in START-BYCC for SBF activation and inactivation. The core components are Swi4 (orange icon), Swi6 (turquoise icon), Whi5 (red icon), the promoter (purple bar), kinases (Cln3, Cln1,2, Clb5,6, Clb1,2) and export protein (Msn5) (white box). The nucleus is represented with a blue-gray background, while the white space corresponds to the cytoplasm. Promoter-bound complexes enclosed in boxes with borders in dark green, light green, and red represent complexes with maximal, residual, and no activity, respectively. White-filled circles represent activatory phosphorylations done by Cln3, Cln1,2, and Clb5,6; whereas the black-filled circles represent the inactivating phosphorylations by Clbs (Clb5,6 & Clb1,2 for Swi6 phosphorylation and Clb1,2 for Swi4 phosphorylation). The Cln (white) phosphorylation on Whi5 and Clb (black) phosphorylation on Swi6 (S160) are needed for export to the cytoplasm. To avoid overcrowding, remaining complexes corresponding to modifications on other free forms are not included in the figure. All the concerned equations are listed in **S2 Text**. Key facts about the abundance, regulation, and localization of all the components are described in **S2 Table**. (B) Ratios of the promoter to Swi4/Mbp1, Swi6, and Whi5 (Roughly based on Ghaemmaghami et al. [84]; e.g., when rescaled, we would have four promoters, 11 each of Swi4/Mbp1, 20 Whi5, and 60 Swi6 molecules).

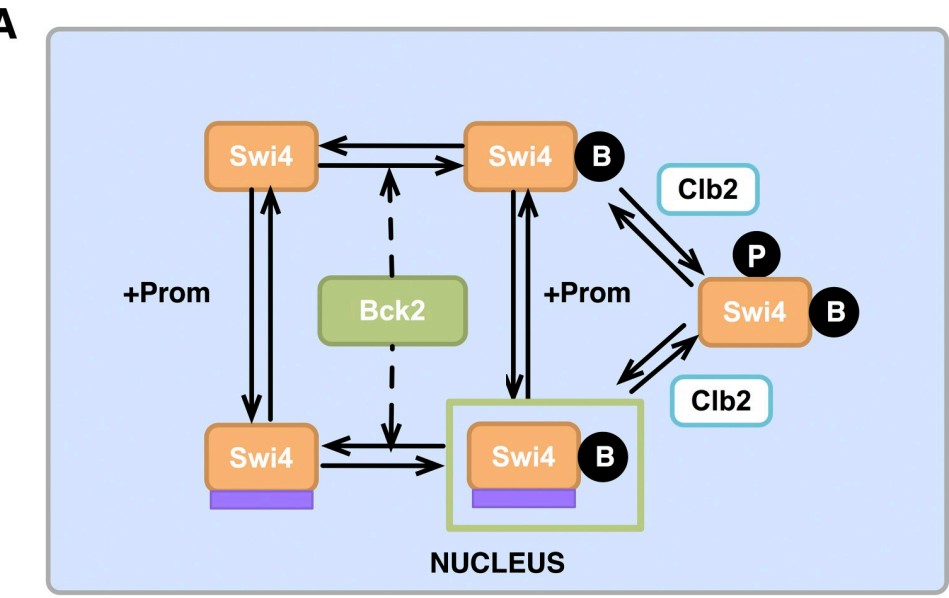

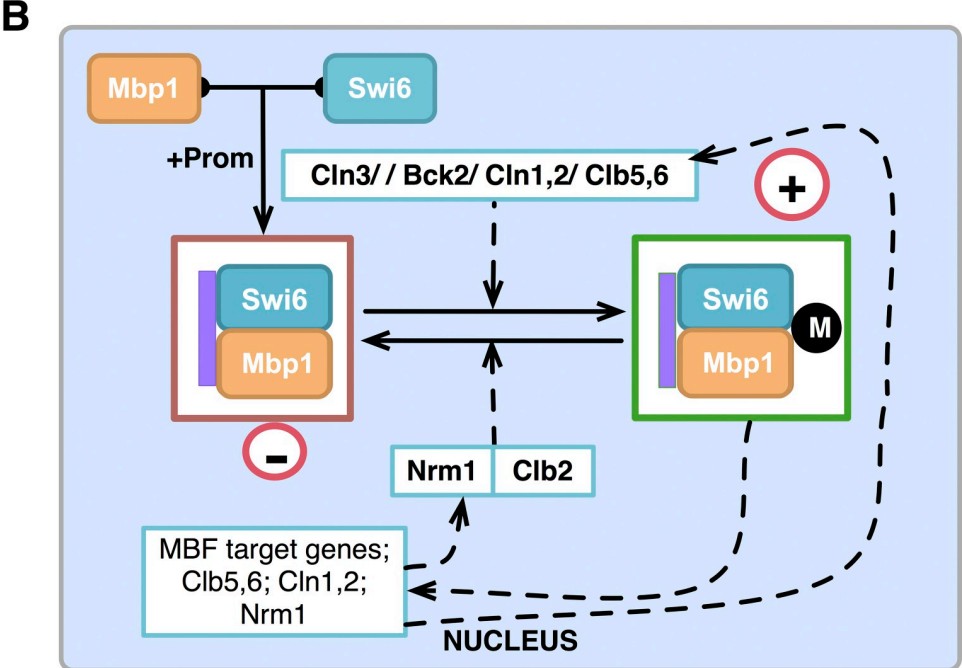

**Fig 3. Other mechanisms involved in the START transition.** (A) Role of Bck2 in the activation of SBF. Bck2 acts on Swi4 and modifies it to a less active form (indicated by the enclosing light green box). The cartoon is a concise representation of regulation by Bck2 (intermediate steps not included). Black-filled circles with 'B' represent the Bck2-induced modification, and active forms are enclosed in a light green box (since these complexes are not as active as Cln-activated forms). Note that only promoter-bound forms are active. Clb2 causes complex inactivation (see equations in **S2 Text**). (B) MBF regulation. MBF alone is inactive in a repressed state (indicated by the enclosing dark red box). It is activated either by cyclins (Cln3, Cln1,2, Clb5,6) or by Bck2 (more active, as indicated by the dark green box). MBF is primarily inactivated by its transcriptional target, Nrm1, resulting in the negative feedback highlighted by the '–' sign. Clb2 is a secondary and minor inhibitor of MBF.

**Table 1. Model Predictions and Validations** (A more exhaustive list of experimentally determined phenotypes is shown in S5 Table). Single letter notations in column one denote: v: validated, p: predicted, and c: contradictory. Grey rows indicate parent groups of mutants, with variations that follow.

| | Mutant | Viability | Functional SBF/MBF complex |
|---|---|---|---|
| v | *cln3Δ mbp1Δ* | **Expt: Viable, large** | **+ SBFa2, SBFa3, SBFa4** |
| v | +*swi6Δ* | **Expt: Viable, large (~swi6del)** | **+ SBFa5, Bck2 activated Swi4 dimer** |
| v | + *mc-BCK2 (5 copies)* | Expt: Viable, large | + SBFa1,4 |
| v | + *whi5Δ* | Expt: Viable, size slightly bigger than whi5-del | + SBFa1 (SBF bound to the promoter), + SBFa2 (SBF phosphorylated on Swi6). |
| v | +*whi5Δ + bck2Δ* | Expt: Still viable, size = bck2del cln3del whi5del = 1.4x WT | SBFa1 (just like the triple Cln3-del bck2-del whi5-del). |
| v | *cln3Δ swi4Δ* | **Expt: Inviable** | **+ Bck2-activated MBF (MBFa)** |
| p | +*whi5Δ* | Pred: viable (large) | Sufficient Bck2 activated MBF (MBFa) |
| p | +*mcBCK2* | Pred: viable | + of MBF and 5X BCK2 is enough to carry on the cell cycle. Bck2-activated MBF. The cell becomes viable in simulation, maybe because enough BCK2 is able to inactivate Sic1 for the cell cycle to go on. |
| p | +*whi5Δ sic1Δ* | Pred: viable and larger than WT. | Bck2 activated MBF (MBFa). Amount decreases as compared to just adding whi5-del. |
| p p | *cln3Δ swi4Δ GAL-BCK2* *cln3Δ swi4Δ whi5Δ GAL-BCK2* | **Pred: rescued** | |
| v | *swi4Δ swi6Δ* | Expt: inviable | – SBFa,–MBFa |
| v | + *SWI6SA4* | Expt: viable | activated MBF, size = swi4del |
| p | + *GAL-CLB5* | Pred: viable, >1x WT | – SBFa/MBFa. CLB5 intact Sic1 |
| p | + *GAL-CLN3* | Pred: not rescued | Need Clb5 to do DNA synthesis |
| p | + *GAL-CLN2* | Pred: rescued, size > 1G | Need Clb5 to do DNA synthesis |
| v | *swi4Δ swi6Δ whi5Δ* | **Expt: inviable** | |
| | *bck2Δ swi6Δ* | **Expt: inviable** | **– SBFa,–MBFa** |
| v | + *SWI6SA4* | Expt: viable | activated MBF (MBFa), size = bck2Δ |
| p | + *GAL-CLB5* | Pred: viable, >1x WT | – SBFa/MBFa. CLB5 intact Sic1 |
| p | +*GAL-CLN3* | Pred: not rescued | Need Clb5 to do DNA synthesis |
| p | + *GAL-CLN2* | Pred: rescued | CKI sufficiently inhibited. |
| v c v p p | *bck2Δ mbp1Δ* *bck2Δ mbp1Δ GAL-WHI5* *mbp1Δ GAL-WHI5* *bck2Δ GAL-WHI5-12A* *mbp1Δ GAL-WHI5-12A* | Viable | |

Mathematical models rely on underlying assumptions for their usefulness and accuracy (S3 Table). To this end, we have highlighted and justified the critical molecular players, mechanisms, and their interactions, along with the detailed wiring, simulated here: http://www.sbmlsimulator.com/simulator/by-start.

## Regulation of START in WT budding yeast cells

**Model description.** In order to understand the role of SBF and MBF in the START transition, we first describe molecular events surrounding the monomers and complexes of these transcription factors, the inhibitor, Whi5, and their target promoters (S1 Fig). Based on known relative protein levels [84] we infer that the monomers Swi6, Whi5, Swi4/Mbp1, and promoter exist in the ratio 3:1:0.55:0.2 (Fig 2B and S2 Table). These monomers bind together rapidly to form complexes based on their stoichiometry and starting concentrations. Due to their high relative levels, Swi6 and Whi5 are available as free molecules in the cell. Most of

Swi4 is bound to Swi6 and Whi5 (*i.e.* most SBF is in the Swi4/Swi6/Whi5 form), and most of Mbp1 is bound to Swi6 forming MBF complexes. Since the promoters are present in limiting levels, most are occupied by their respective transcription factors, Swi4/Swi6/Whi5 (inactive SBF) or Mbp1/Swi6 (inactive MBF). The remaining SBF and MBF complexes are free and unbound to promoters. This sets the stage to discuss the various regulatory events that ensue during the START transition.

**SBF activation by Cln kinases in the late G1 phase.**    Since a transcription factor complex can be transcriptionally 'active' only when bound to a promoter, for the following discussion, we focus on promoter-bound SBF complexes and describe their activation by Cln kinases and inactivation by Clb kinases. We assume that a similar kind of regulation occurs on the promoter-free complexes.

When the yeast cell is in G1, most of the promoter-bound SBF complexes are in an inactive, Whi5-bound state. When Cln3 accumulates in late G1, Cln3/CDK activates SBF by phosphorylating Whi5 and Swi6 at several residues [47,48], resulting in the doubly phosphorylated form. We assume that this form is unstable and dissociates into phosphorylated SBF and phosphorylated Whi5 (**Figs 2 and S2**). Aided by the karyopherin export protein Msn5, phosphorylated Whi5 subsequently moves to the cytoplasm (and stays there until mitotic exit/FINISH transition) [57,60], leaving SBF transcriptionally active.

**SBF inactivation in the late S phase.**    In late S phase, SBF is inactivated by a second round of phosphorylation by mitotic cyclins (Clb1,2; Clb6) [50] (**Figs 2 and S2**). Presumably, these phosphorylations occur on Swi4 [31] and Swi6 (on residue S160, different from those targeted by Cln3) [59], leading to dissociation of the complex from the promoter and cytoplasmic localization of Swi6 (facilitated by Msn5 [85]).

The START-BYCC model explicitly includes the discrete steps by which important active forms of SBF (enclosed in dark green boxes) are inactivated and exported to the cytoplasm (**Figs 2 and S3**). This series of molecular events comprises a detailed consideration of intermediate active complexes that we do not expect to see in significant portions in wildtype cells. However (as described in later sections), these considerations become helpful in understanding the regulatory logic underlying non-phosphorylable mutant phenotypes. Thus, following Clb phosphorylation on the Swi4 moiety of SBF, each of the intermediate SBF complexes dissociates from the promoter, turning off SBF-regulated genes. Then, the free SBF forms (promoter-unbound) move to the cytoplasm, aided by Msn5. We assume that these complexes dissociate once in the cytoplasm and remain there until their corresponding phosphatases reverse their modifications.

**Activation of SBF by Bck2.**    In addition to activation by cyclins, SBF is also activated by Bck2 in late G1 [49]. The viability of *cln3Δ* and the inviability of the double mutant *cln3Δ bck2Δ* [49] emphasize the importance of Bck2 in SBF activation. Bck2's mechanism of action is not very clear, but it is thought to be independent of CDK phosphorylation and Whi5 [47,49]. In START-BYCC, we consider a mechanism by which Bck2 modifies SBF to alternate active forms (**Fig 3A**). We determine the relative contributions of Cln3 and Bck2 to SBF/MBF activation from the relative sizes of cln3Δ and bck2Δ mutants (*cln3Δ >> bck2Δ >>* WT; see the section on the role of Bck2[49,86]). We assume that the complexes activated by Bck2 are less active than the Cln-activated forms and indicate this with light green outlined boxes (**Fig 3A**).

In addition to the Swi4/Swi6 heterodimer of SBF, we assume that (in the absence of Swi6) Swi4 too, has some residual activity and that Bck2 activates homodimeric Swi4. This assumption becomes important in the context of mutants *swi6Δ*, *swi4Δ swi6Δ*, and *swi6Δ* in the genetic background of *cln3Δ* or *bck2Δ* (described below).

**Regulation of MBF.**    Alongside SBF, the other important transcription factor operating at START is MBF, a heterodimer of Mbp1 and Swi6. The primary cell cycle targets of MBF

considered in START-BYCC are Clb5,6. Due to the functional overlap between SBF and MBF [26], we assume, in START-BYCC, that MBF can also activate Cln1,2 (targets of SBF) and that SBF can activate Clb5,6 (targets of MBF).

MBF is activated by both Cln3 and Bck2 by independent mechanisms and inactivated by Clb2 and by Nrm1, one of MBF's targets [87] + Cln2 and Clb5 activate MBF, albeit to a lesser extent. In START-BYCC, we have incorporated the positive and negative feedback loops for the activation and inactivation of MBF, as depicted in **Fig 3B**.

## Wildtype simulations of the budding yeast cell cycle

The various mechanisms described thus far (**Figs 2 and 3**) correspond to the typical set of biomolecular interactions that occur in a wildtype cell through the cell cycle. In START-BYCC, we have incorporated several intermediate reactions that could potentially occur only in specific overexpression or knockout mutants. For example, **S3 Fig** considers intermediates that we expect to observe only in non-phosphorylable mutants. The model also considers different pools of complexes (free and promoter-bound) and monomers, and mechanisms for their modification and localization (**S2 Table**). The levels (and significance) of each of these complexes/intermediates depend on the amounts of starting monomers, the nature of the mutant, and the growth rate (reflected in the mass doubling time).

We convert this entire picture [17] (highlighted in **Figs 2, 3, and S3**) into ordinary differential equations, each reflecting the dynamic fate of one biomolecular entity (variable). These equations are then used to run time-course simulations of the system (solved numerically with the equations, parameters, and initial conditions; details provided in **S2 Text;** http://www.sbmlsimulator.com/simulator/by-start). Results from a typical time-course simulation of a wildtype cell are shown in **Fig 4**. Each graph tracks the concentration of different sets of variables in normalized units spanning 300 min (~3 cell cycles), along with the 'mass' variable acting as a proxy for cell size. By default, cells grow in glucose with a mass doubling time of 90 min (in reasonable agreement with Brewster 1994 [88]), and the daughter cells are tracked (mass at division = ~0.46 of total daughter+parent mass pre-division). The total cycle time for the daughter corresponds to 107 min, again reflecting experimental timings [57], but daughter cell G1 length is prolonged compared to experimental observations at ~50 min.

Our current expanded model captures the cellular dynamics of cyclins, cyclin antagonists, transcription factor complexes, and checkpoint proteins (**Fig 4A–4D**), similar to the BYCC model [17]. Wildtype cells start in G1, with G1 stabilizers active (high Sic1, Cdc6, Cdh1). In late G1, as cells reach a size threshold to trigger the START transition, a nuclear build-up of G1/S activators Cln3 and Bck2 results in the activation of SBF, MBF, and subsequently, Cln1,2 and Clb5,6. The activation of the feedback between the cyclins and SBF/MBF ensures an irreversible START transition [89,90] (when not under starvation). The nuclear export of phosphorylated Whi5 is another hallmark of SBF activation (in accordance with Di Talia 2007 [57]). In turn, the S phase cyclins (Cln1,2; Clb5,6) lead to concurrent initiation of DNA synthesis (origin of replication, ORI) and bud formation (BUD), as observed experimentally. We set the firing of the origin, ORI = 1, as the marker for START. Cln3 remains nuclear until late G1 and moves to the cytoplasm in late G2 (as observed by Verges et al., 2007 [83]).

Transcriptional activation of S phase cyclins by SBF and MBF causes CKI degradation (by Cln1,2) and Cdh1 inactivation (by Clb5,6). This results in the accumulation of mitotic cyclins, Clb1,2 (further aided by the Clb1,2, Mcm1 feedback), and entry into the M phase. Following this transition, sister chromatids align at the metaphase plate, with ensuing Cdc20 and Cdc14 activation resulting in exit from mitosis. Since the current model focuses on START, we use the simplified, older version of BYCC to emulate the mitotic exit and FINISH transition.

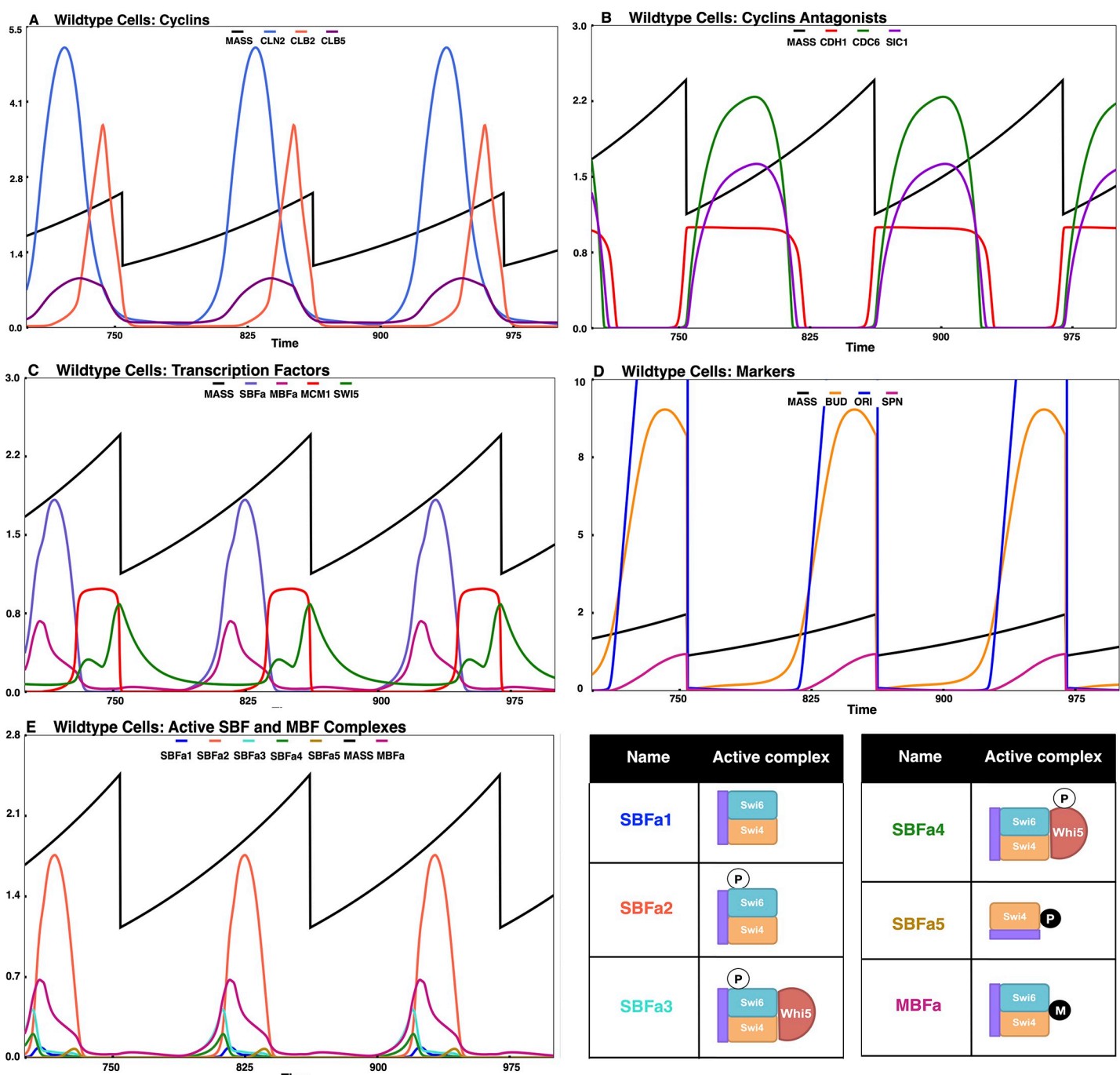

**Fig 4. Simulation of wildtype cells in glucose.** Each panel tracks the following proteins/components for ~3 cell cycles in glucose with mass doubling time 90 min and daughter cycle time ~107 min. Mass acts as a proxy for the growing and dividing cell; here we follow the daughter cells (~0.46 x total_size_at_division; see *Equations in S2 Text*). (A) Cyclins (Cln2, Clb5 (active), Clb2 (active)), (B) Cyclin antagonists (Cdh1, Sic1 (active), Cdc6 (active)), (C) Transcription factors (SBF, MBF, MCM1, Swi5), (D) Markers (CDK targets (BUD, ORI, SPN) used to indicate the occurrence of physiological events via concentration threshold (BUD = 1 (bud emergence), ORI = 1 (DNA synthesis initiation), SPN = 1 (spindle alignment in metaphase)), (E) Active SBF and MBF complexes. Each form shown (SBFa1-a5, MBFacln, MBFabck) is the activity of SBF or MBF contributed by a particular form. The complexes contributing to the different forms (all bound to promoter) are as follows: SBFa1 = unmodified SBF (SBFB in the model); SBFa2 = SBF phosphorylated on Swi6 (SBFB6P+SBFB6PQ); SBFa3 = SBF-Whi5 complex phosphorylated on Swi6 (WSB6P+WSB6PQ); SBFa4 = SBF-Whi5 complex phosphorylated on Whi5 (WSB5P); SBFa5 = Swi4dimers activated by Bck2 (Swi4B); MBFa = MBF activated by Clns or Bck2. Check and cross marks denote the presence or absence (due to mutation) of specific complexes. The absence of any sign denotes that the specified complex is absent in the simulation of that strain. The black curve in all panels denotes mass (corresponding to exponential cell growth and division). As shown in (E), for wildtype cells, the dominant form of SBF is SBFa2, the form of SBF phosphorylated on Swi6, whereas the dominant form of MBF is the Cln-activated MBF.

Incorporation of our detailed START model with our more recent detailed FINISH models [41], and the relevant detailed descriptions of FEAR and MEN pathways and the spindle position checkpoint, are beyond the scope of this work [91,92].

In line with the expansion of the molecular mechanism underlying START, we track not only the total complexes, but the distribution between various active forms of SBF and MBF (**Fig 4E**). In later sections, we present simulations of mutant cell types similar to the wildtype plot of SBF/MBF complexes in **Fig 4E**. For viable cells, we show cycles after the cell has reached a steady state. The cell size at division in our simulations roughly corresponds to the mean cell volume determined in experiments. In addition to quantitatively measuring cell mass from simulations, we can also deduce a qualitative size for different mutant cell types by comparing the amounts of their active SBF and MBF complexes (icon table beside **Fig 4E** and other mutant simulations). In all cases (wildtype and other mutant phenotypes), we report the observed phenotypic size of the mutants as fold-change relative to wildtype cells in glucose.

## Size control

Nutrient modulation of size control represents a key aspect of the budding yeast cell cycle [7], wherein budding yeast exhibits a thresholding effect – i.e, cells must grow to a critical size to initiate budding and S phase. This control is achieved through tight coupling between cell growth and division. We, therefore, revise the molecular details of the size control mechanism in START-BYCC for a more accurate depiction of size control and its coupling to the cell cycle.

## Cln3 activation of SBF, MBF

Cln3 is implicated in size control, since mutants with Cln3 deletion or increased Cln3 cytoplasmic export are larger than WT [93,94] and mutants overexpressing Cln3 or with increased Cln3 nuclear import are smaller than WT. Cln3 was thus thought to be a nuclear sensor of cell size, triggering START only when the cell reached the threshold size. Although Cln3 has been identified as an important regulator of START, Cln3 knockout mutants are still viable. This is due to the activity of Bck2, which is activated by glucose and known to promote START [49].

The earlier models of the cell cycle, including BYCC [16,17], incorporate the role of Cln3 in sensing size in a direct mass-dependent manner: as cell volume grows, the total Cln3 protein level in the cell increases in parallel, with its concentration remaining constant. As its level rises, Cln3 migrates to the nucleus and concentrates there. Assuming the nuclear volume does not change significantly, the nuclear concentration of Cln3 grows proportionally to cell size. When the cell reaches a threshold size, this mass-dependent nuclear accumulation of Cln3 (& Bck2) triggers START by activating SBF abruptly (modeled with a classic ultrasensitive Goldbeter-Koshland GK switch) [69]. Thus, the START transition behaves like a switch in the BYCC model [16,17]. Recent experiments, however, suggest that the nucleus grows proportionally to the cell during the cell cycle, challenging this hypothesis [66,95].

## Size control in START-BYCC

We treat cell size control as the outcome of actively regulated nuclear import of Cln3 (and Bck2) prior to START. Recent experiments show that, in early G1, the Cln3/Cdc28 complex is sequestered to the endoplasmic reticulum (ER) membrane by Ssa1/2, Whi3, and other negative regulators of START. The J-chaperone protein Ydj1 was originally reported to be involved in phosphorylation of Cln3 through CDK to mark it for degradation, but a *ydj1* mutant was shown to result in large cell size, suggesting it may play additional roles in Cln3's function [96,97]. It has since been shown that Ydj1 contributes to release of Cln3 from the endoplasmic

reticulum and nuclear localization, and available Ydj1 is tied to cellular growth rate, implying Ydj1 could act as a cell growth rate sensor that helps gate START initiation [83].

The START-BYCC model assumes that Ydj1, in response to increasing mass, moves Cln3 from ER into the nucleus abruptly in late G1. Nuclear Cln3 phosphorylates and inactivates Whi5, resulting in the nuclear export of Whi5 and activation of SBF [47,48]. Since Ssa1 is the protein that retains (and thus inhibits) Cln3 in the ER, we assume that Ssa1 inactivates Cln3 [83]. In order to explain the observation that Cln3 is nuclear from late G1 to late S/G2, we propose that Ssa1 is activated by Clb2 and Swi5 (**S4 Fig**). Recent work has identified Mad3 as an additional timer mechanism [71]. As Cln3 levels increase over G1, it was found that SCF-mediated Cln3 degradation diminishes as Mad3 is itself degraded by APC [71]. In *Δmad3* cells, nuclear Cln3 is unvarying across G1, but Whi5 dilution remains as a separate cell sizing mechanism meaning Mad3 mutants produce only modestly smaller cells [64,71]. In our model, these indirect late mitotic effects resulting in Cln3 degradation/inactivation also go through the Ssa1. Since little is known about the regulation of Bck2, we assume that its regulation is similar to that of Cln3. We have modified the size control mechanism from the BYCC model to reflect these hypotheses and assumptions but retain Cln3 and Bck2 as the ultimate size sensors (albeit through the proxy of growth rate-based Ydj1/Ssa1 [11,57,83,98]).

Results from simulations of START-BYCC compare well with observations from a classic experiment in size control measuring cell size distribution for different nutrient media and growth rates [99]. Our simulation results show that we have the same proportional changes in mass and median volume (mass, our cell size measure, exponentially increases from 1.5 to 2.5 a. u. aligning with the increases of the median volume from 15 to 25 $\mu m^3$, reported in Lord and Wheals, 1980) (**Fig 5**). Slower growth rates result in smaller cell volumes (taken to be mass at division in START-BYCC simulations) and higher growth rates (richer nutrient media) result in higher median cell volumes. Furthermore, we show that START-BYCC reproduces the observation that daughter cells display a progressive G1 delay (activation of ORI/BUD in the model) in response to slower growth rates (poorer nutrient media) that is characteristic of strong size control [99]. Also, in line with our expectations, our simulation shows that the longer cycle times in slower growth media occur mostly due to a delay in G1 (wherein smaller, slower-growing cells wait to reach the size threshold), leaving the budded phase almost constant (**S5 Fig**).

## Export and localization of START proteins

Several studies on START transition have emphasized the crucial role of localization of the core proteins (in their various modified/complexed forms) [57,59,60,81]. We consider the export and localization of the START monomers by building a compartmentalized model (nucleus, cytoplasm inside a single cell). The ratio of the sizes of the nucleus and cytoplasm is predefined (0.2:0.8 in our case) and remains constant throughout the cell cycle [95]. We detail the specifics of the localization of Whi5, Swi6, and Swi4 below, and summarize them in **S6 Fig**.

## Whi5 localization

Single-cell experiments reveal Whi5 is in the nucleus only in G1, moving to the cytoplasm with the onset of the START transition and remaining there until the next cycle begins [57]. We have encoded this behavior in the model by allowing phosphorylation of all forms of Whi5 in the nucleus by G1 and G1/S cyclins, Cln3, Cln1,2, and Clb5,6. These phosphorylation events signal both free and SBF-bound Whi5P to be exported to the cytoplasm in late G1 by the transport protein Msn5 (**S6 Fig**). Additionally, any Whi5 bound to SBF with Swi6 phosphorylated by Clb kinase (Q-form; see below) is also exported to the cytoplasm. Cytoplasmic Whi5P is then dephosphorylated by Cdc14, which is activated at mitotic exit (**S6 Fig, Step 4**).

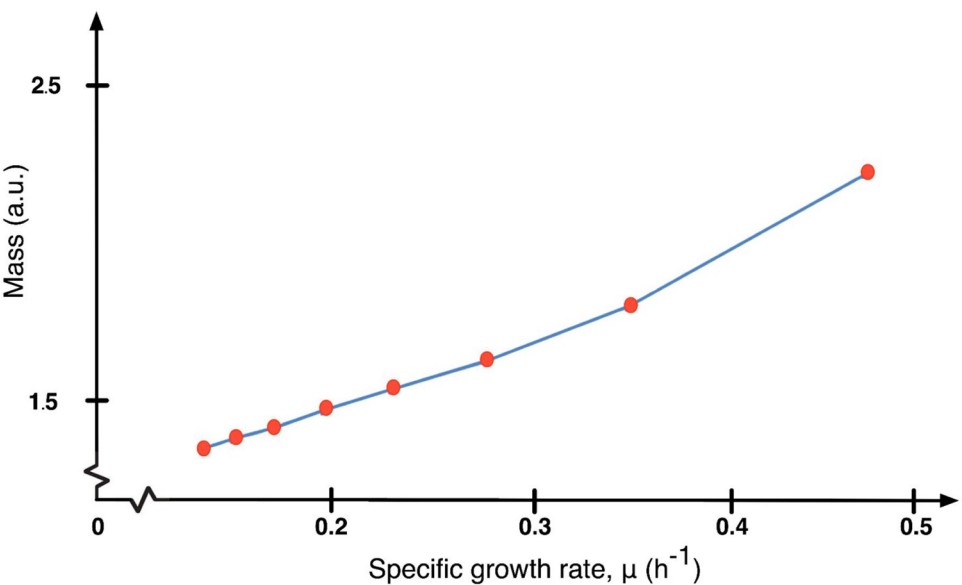

**Fig 5. Cell size as a function of growth rate in budding yeast.** The blue curve connecting red circles shows cell size (mass) at division at specific mass doubling times (MDT) from 90–300 min (30 min interval) and corresponding specific growth rates (ln2/MDT) from START-BYCC simulations. This curve shows the same trend as the experimental graph in Lord and Wheals (1980), where median cell volumes for populations are plotted at different growth rates.

## Swi6 localization

Swi6 localization is known to depend on the serine residue S160, which, when phosphorylated in late S-phase by Clb6, allows export to the cytoplasm [59]. In the Swi6 S160A mutant (serine mutated to alanine), Swi6 remains in the nucleus throughout the cell cycle. START-BYCC designates this S160 phosphorylated form as the 'Q-form' (black phosphorylation site in **S6 Fig**) to distinguish it from the other activatory phosphorylations on Swi6 by Cln kinases (designated 'P-forms'; white phosphorylation site in **S6 Fig**). In addition to Clb5,6 [50], we assume that Clb1,2 can also bring about the S160 phosphorylation (Q-form). Swi6 in its Q-form is exported to the cytoplasm (in the presence of Swi4, aided by Msn5) and stays there from the late S phase through the end of the cycle [85]. In START-BYCC, the phosphatase, PP2A dephosphorylates the P-form (**S6 Fig, step 3**). Similar to Whi5, we assume that the final dephosphorylation (on Q-form) of Swi6 happens at the mitotic exit by Cdc14, followed by quick re-import [50,100] (**S6 Fig, step 4**).

## Swi4 localization

Unlike Whi5 and Swi6, Swi4 shows nuclear localization throughout the cell cycle [81]. For this reason, we assume that phosphorylated Swi4 (irrespective of the compartment) is dephosphorylated by an unspecified active phosphatase (Ppase in our case) without allowing it to wait for Cdc14 to accumulate at mitotic exit (**S6 Fig, step 3**).

## Simulation of localization and export of monomers

Through numerical simulations based on the aforementioned formalization, we have captured nearly all localization events of monomers during different phases of the cell cycle (**Fig 6**). Firstly, we examine the profiles of phosphorylated and cytoplasmic monomeric components and the timing of their nuclear export. We observe that phosphorylated Whi5 is exported to

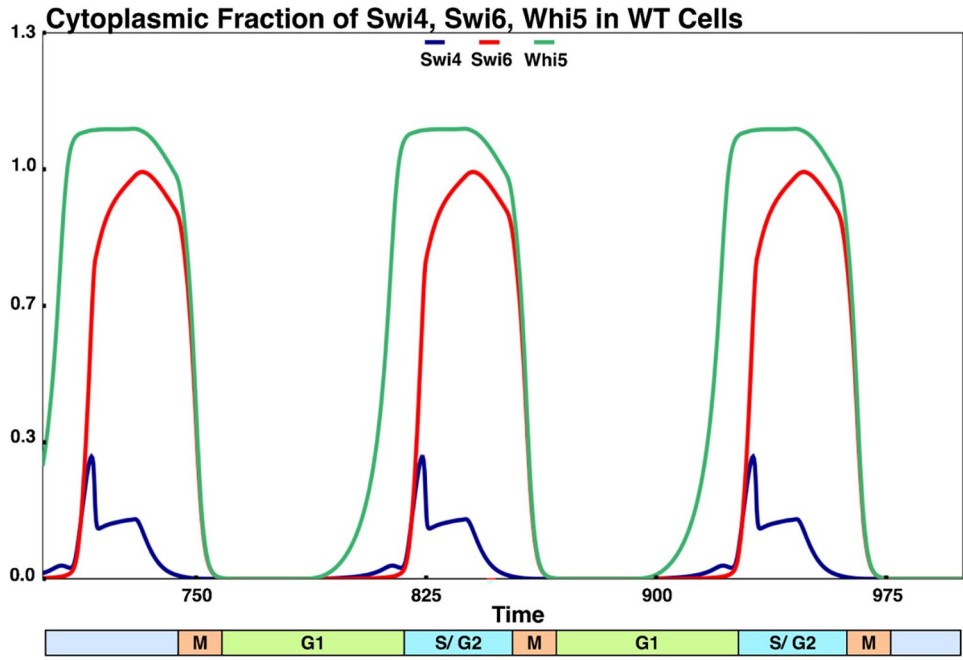

**Fig 6. Simulation of the timing of localization (export) of different monomers.** The cytoplasmic fraction of monomers Swi4, Swi6 and Whi5 are plotted against time. The bar at the bottom of the graph shows timing w.r.t. phases of the cell cycle. In the model, the onset of S and M phases correspond to DNA synthesis (ORI = 1) and spindle assembly checkpoint (SPN = 1), respectively. In compliance with experiments, Whi5 enters the cytoplasm (exits the nucleus) in late G1, followed by Swi6 in S-phase. Both Swi6 and Whi5 are cytoplasmic until mitotic exit. Swi4 is mostly nuclear at all times. Details are discussed in the main text.

the cytoplasm in late G1, while phosphorylated Swi6 is exported after START (late S phase). Both monomers stay in the cytoplasm until mitotic exit (**Fig 6**). There is a small level (<10%) of phosphorylated Swi4 in the cytoplasm in S/G2 phases of the cell cycle. These simulation results correspond well with previously discussed experimental findings on the localization of Whi5, Swi6, and Swi4. We observe that most of Swi6 goes to the cytoplasm in late S phase, following Whi5. This is because Swi6 constituting MBF, and certain phosphorylated forms of SBF are incapable of moving to the cytoplasm in START-BYCC. Only the doubly phosphorylated Swi6 forms are capable of cytoplasmic export (**Figs 2 and S6**).

Secondly, we are also able to reproduce the results pertaining to the importance of the export protein Msn5 in the cell. *msn5Δ* mutants are known to be larger than WT cells [85], and our simulations show that these cells are indeed considerably larger, since SBF localization is upset and there is only active MBF (**S7 Fig**). In *msn5Δ*, we observe that both Whi5 and Swi6 are nuclear at all times, in accordance with experiments.

### Timing of individual cell cycle phases

While our time-course predictions closely recapitulate overall cell cycle timing for mother and daughter cells, the length for specific phases deviates from experimental observations [57,88,99]. Given that our model focuses on the START transition, while including only a key representative set of non-START cell cycle factors, it is not surprising that the details of timing differ. Nevertheless, we do note that despite minor inconsistencies in the predicted timings for individual cell cycle phases, the phenotypic predictions of a vast majority of mutants are captured qualitatively and quantitatively, with further experimental corroboration as well [101].

In summary, for wildtype budding yeast cells, our current START-BYCC model can explain the dynamics and underlying mechanisms by recapitulating known experimental phenotypes, from START and the rest of the cell cycle. Next, we discuss extending our model to explain mutant phenotypes.

### Case of the non-phosphorylable mutants

We use the START-BYCC model to describe several mutant phenotypes, most of which are implicated in the START transition. Since a critical gap in our previous understanding of START is the complex role of phosphorylation in regulating the transition, we start with the detailed characterization of the non-phosphorylable START mutants: single mutants *WHI5-12A*, *SWI6-SA4*, and the double mutant *WHI5-12A SWI6-SA4*. Recent studies have shown that there is a delay in the START transition only when neither Whi5 nor Swi6 can be phosphorylated [59,60]. This is contrary to the previous belief that Whi5 phosphorylation is essential for relieving the inhibition of SBF [17,47,48].

The START-BYCC model accommodates these new findings by considering all possible phosphorylation states and intermediate complexes of Whi5, SBF, and the target promoter (**S8 Fig**). In the mutant *WHI5-12A*, Whi5 cannot be phosphorylated, but the phosphorylation sites on Swi6 are intact (**S8 Fig**). Therefore, only Swi6 gets phosphorylated in both SBF and SBF-Whi5 complexes to yield the active transcription factor (**S8A Fig**). On the other hand, in the *SWI6-SA4* mutant, Swi6 cannot be phosphorylated, but the phosphorylation sites on Whi5 are intact. Therefore, in this mutant, the transcription factor complex can be phosphorylated only in its Whi5-bound form (**S8B Fig**). In both single mutants, we assume that the phosphorylated transcription factor complexes are fully active (akin to the complexes present in wildtype cells) so that there is no difference in size (**Fig 7A–7B**).

In the double mutant *WHI5-12A SWI6-SA4*, neither Swi6 nor Whi5 can be phosphorylated (**S8C Fig**). Consequently, the SBF-Whi5 complex remains unphosphorylated and inactive. Even though promoter-bound unmodified SBF has some residual activity and Bck2 is still present, excess Whi5 renders SBF inactive (**S8C Fig**). So, in the double mutant, SBF is completely off. Despite the absence of active SBF, these cells are still viable, albeit large (since nuclear Whi5-12A can inhibit MBF, and Swi4 can compete with Mbp1 for Swi6). The viability of *WHI5-12A SWI6-SA4* can be explained by the presence of the almost intact MBF that can transcribe Clb5,6 and Cln1,2 (due to functional overlap with SBF; **Fig 7C**). The double mutant *GAL-WHI5-12A SWI6-SA4* is inviable because the excess Whi5 can inhibit the activity of MBF (**Fig 8J**), in line with experimental observations [47,48]. In START-BYCC, Whi5-12A is a stronger inhibitor of SBF than wildtype Whi5 due to its nuclear localization. Similarly, Swi6-SA4 makes SBF weaker than wildtype Swi6 (weaker form dominates), as observed in the difference in size between *bck2Δ* and *bck2Δ swi6Δ SWI6-SA4* (slightly larger) (**Fig 8B and 8O**).

In summary, START-BYCC and simulations explain in detail the role and working of these non-phosphorylable mutants by replicating the experimental findings with the sizes of *WHI5-12A* (**Fig 7A**) and *SWI6-SA4* (**Fig 7B**) being comparable to WT, but the double mutant significantly larger (**Fig 7C**).

### Simulation results of mutant phenotypes

Using our current model of the cell cycle, we have simulated several mutants pertaining to START and other phases of the cell cycle (as per **S5 Table** and **Fig 8**). We have summarized our results in **S5 Table** where we list the mutants, their phenotypes as observed in experiments, and the results from the simulations of our current model. Our model predictions agree with all the known expected phenotypes, and some have been experimentally validated [101].

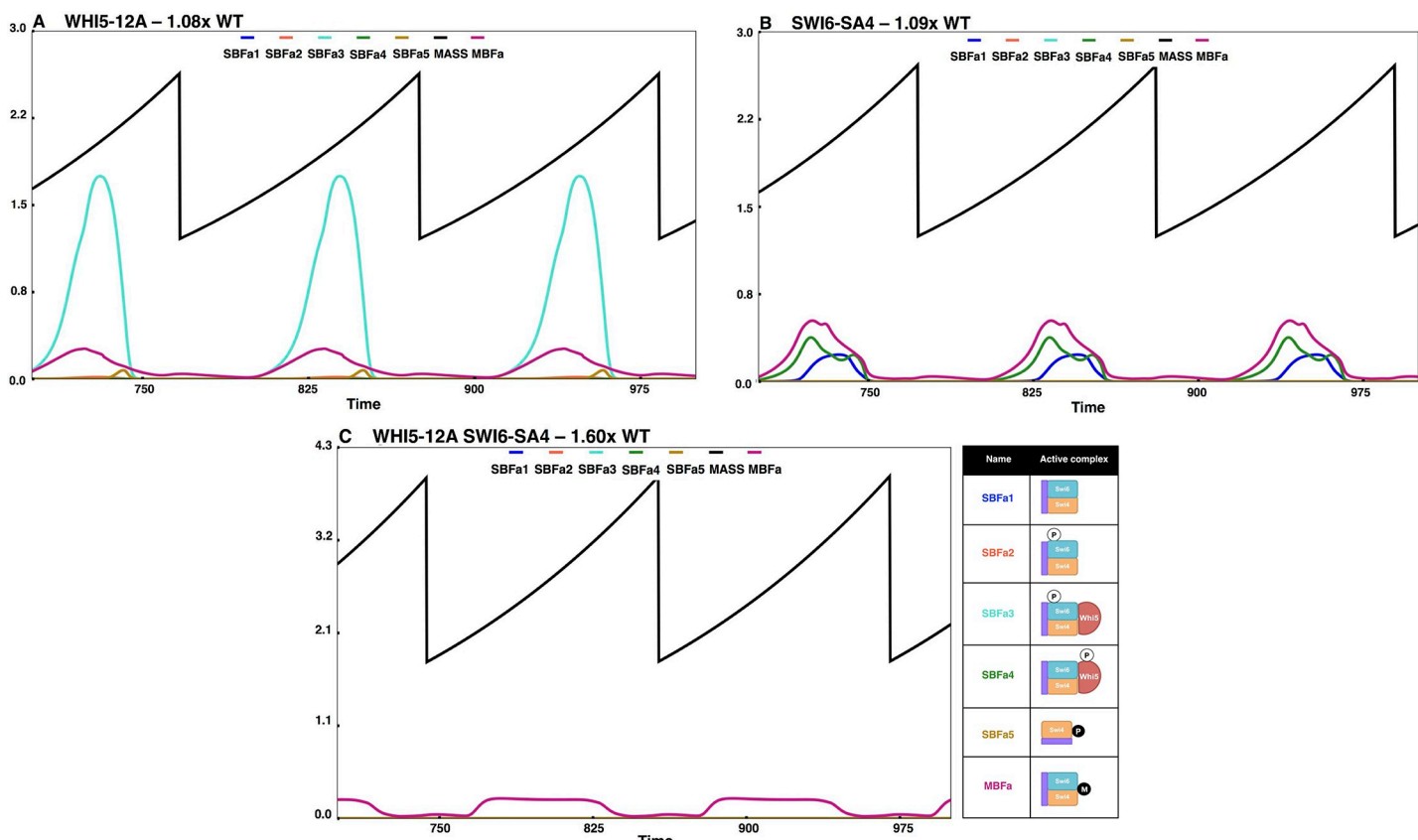

**Fig 7. Simulation of non-phosphorylable mutants.** The simulations shown here track the cell size and key active SBF/MBF fractions, in specific non-phosphorylable START mutants, and the descriptions are as in Fig 4. (A) *WHI5-12A* (SBF-Whi5 complex phosphorylated on Swi6 and MBF are the primary active forms–SBFa3, MBFa; cells are ~WT size), (B) *SWI6-SA4* (SBF activated by Bck2, SBF-Whi5 complex phosphorylated on Whi5, and MBF are the primary active forms–SBFa4, MBFa; cells are ~WT size), (C) *WHI5-12A SWI6-SA4* (only very little of Bck2 activated MBF forms are present–MBFa; cells are larger than WT).

In the following subsections, we present simulations of a few important START mutants that highlight the roles Bck2, Cln3, and the transcription factor monomers Mbp1, Swi6, and Swi4. In each simulation plot corresponding to the mutants, we have included iconic representations (cartoon depictions) of the major active SBF and MBF complexes to follow the relative abundances of these complexes and the resulting mutant phenotypes. We also indicate the names of these complexes below.

## Mutants pertaining to the role of Bck2

The role of Bck2 in START is not well understood. We, therefore, use START-BYCC to simulate mutants of Bck2 and Cln3. In the *cln3Δ* mutant, most SBF complexes are bound to Whi5 and are inactive, with very little SBF left for activation by Bck2 (there is very little SBFa1 and SBFa5). As for MBF, only the Bck2-activated form is present, and the remaining MBF units are inactive. Therefore, *cln3Δ* mutants attain a very large size to accumulate enough Bck2 to sequester SBF to an active form (from Whi5) to trigger START[86] (**Fig 8A**).

In contrast to the *cln3Δ* mutant, in the *bck2Δ* mutant, most of the active SBF complexes are intact, and only the Swi4B (SBFa5) forms remain (**Fig 8B**). These cells are therefore only slightly larger than wildtype cells (1.32x WT; **Fig 8B**). Note that the active SBF/MBF complexes responsible for START in *cln3Δ* and *bck2Δ* mutants complement each other. Hence, for *bck2Δ*

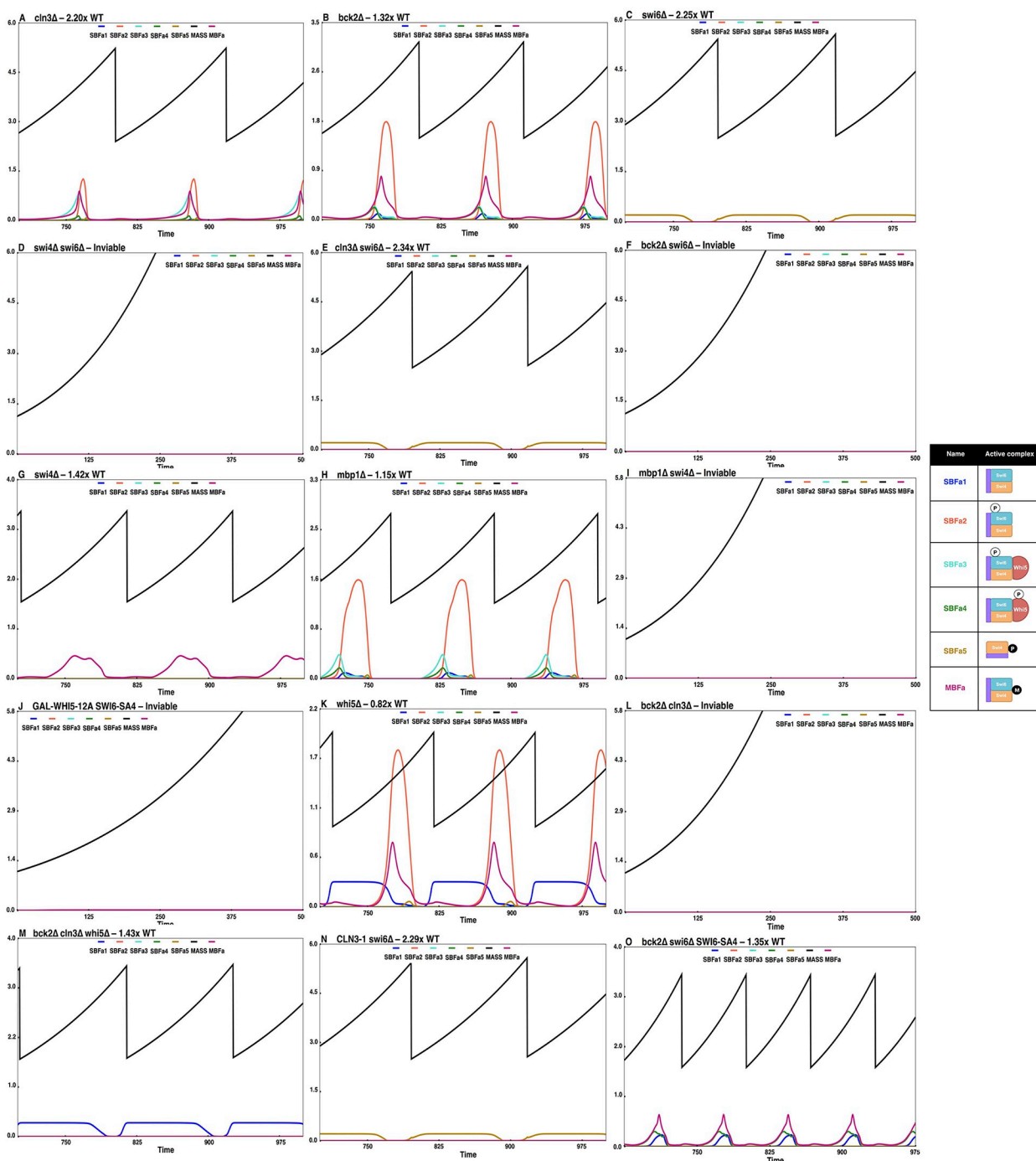

**Fig 8. Simulation results of a few important START mutants.** Simulations of the following mutants and their steady state sizes (viability/inviability) and the active SBF/MBF complexes present are shown: (A) *cln3Δ* (only Bck2 activated forms are present; cells are very large), (B) *bck2Δ* (only Cln-activated forms are present; cells are slightly larger than WT), (C) *swi6Δ* (only Swi4dimers (SBFa5) present; cells are viable yet large), (D) *swi4Δ swi6Δ* (no SBF/MBF; cells are inviable), and (E) *cln3Δ swi6Δ* (only Swi4dimers (SBFa5) present; cells are viable yet large), (F) *bck2Δ swi6Δ* (no active SBF/MBF; cells are inviable), (G) *swi4Δ* (only MBF present; cells are very large), (H) *mbp1Δ* (only SBF present; cells are slightly larger than WT), (I) *swi4Δ mbp1Δ* (no SBF or MBF; cells are inviable), (J) *GAL-WHI5-12A SWI6-SA4* (excess non-phosphorylable Whi5 inhibits MBF; cells are inviable), (K) *whi5Δ* (cells begin the cycle with less active SBF (SBFa1) instead of Whi5-bound inactive SBF, and get converted into more active forms by Clns (SBFa3). Active MBF is also present. Hence, cells are smaller than WT.), (L) *bck2Δ cln3Δ* (no active SBF/MBF; cells are inviable), (M) *bck2Δ cln3Δ whi5Δ* (deletion of Whi5 relieves SBF and the unmodified form of SBF (SBFa1) is consistently present and cells become viable), (N) *CLN3-1 swi6Δ* (only Swi4dimers (SBFa5) present; cells are viable yet large), (O) bck2Δ swi6Δ-SA4 (SBFa1, SBFa4, and MBFa present; cells are slightly larger than WT).

to show a significant increase in size (as observed in experiments; 1.3x WT [37]), the contribution from Bck2-activated forms should be significant in wildtype cells. This would automatically result in a smaller size for *cln3Δ* mutants. Therefore, ensuring a large size for *cln3Δ* in our simulations (to match 1.8–2.7x WT [35,51]) would make *bck2Δ* larger than wildtype.

Besides SBF and MBF, in START-BYCC, we assume that homodimeric Swi4 has residual transcriptional activity. This assumption is consistent with the following mutant phenotypes: *swi6Δ* is viable and large, whereas *swi6Δ swi4Δ* is inviable [23]) (**Fig 8C–8D**). The activation of Swi4 dimers is then brought into question. Here, while the deletion of Cln3 in *swi6Δ* cells has no deleterious effect on their viability [23] (**Fig 8E**), the mutant *swi6Δ bck2Δ* is inviable [61] (**Fig 8F**). These results suggest that the Swi4 dimers depend on Bck2 for their activation (resulting in Swi4B and SBFa5 in the model; **Fig 3A**).

## Mutants pertaining to regulation of MBF

The transcription factor MBF has a large functional overlap with SBF [26]. In START-BYCC, we calibrate the relative importance of SBF and MBF, and their functional overlap based on the relative sizes of several known single and double deletion mutants of Mbp1, Swi4, and Swi6 (**S5 Table**). For instance, *swi4Δ* cells (absence of SBF, presence of MBF) are about 1.3–1.5x WT size (**Fig 8G**), whereas *mbp1Δ* cells (absence of MBF, presence of SBF) are approximately 1.2–1.3x WT size [21] (**Fig 8H**). The double mutant is inviable, signifying that either SBF or MBF should be present for the activation of START and the viability of cells (**Fig 8I**).

To explain the experimentally determined phenotypes of *GAL-WHI5* mutants in START--BYCC, we consider that Whi5 inhibits MBF through non-specific binding in the presence of excess Whi5 (**Fig 3B**). For example, the lethality of the mutant *GAL-WHI5-12A SWI6-SA4* relies on consideration of this inhibition [47,60]. The double mutant *WHI5-12A SWI6-SA4* is viable and large solely due to the presence of MBF (MBFa in **Fig 7**) and to a lesser extent due to SBF activated by Bck2 (SBFa2) (**Fig 7C**). However, the mutant *GAL-WHI5-12A SWI6-SA4* is inviable [60]. While Whi5 binding only to SBF does not explain this scenario, assuming that excess Whi5 can bind to and inhibit MBF accounts for the inviability of *GAL-WHI5-12A SWI6-SA4* mutant (**Fig 8J**). In vitro studies support this hypothesis of Whi5 inhibition of MBF [47].

## Mutants pertaining to the interplay between Whi5, Cln3, and Bck2

Next, we survey the interplay between the activators, Cln3 and Bck2, and inhibitor, Whi5, through a set of relevant mutants. The *whi5Δ* mutant is small in size [47,48] because, in the absence of Whi5, the SBF complex starts G1 in its uninhibited form, which still carries some residual activity. Thus, in *whi5Δ*, the cells need not wait for the accumulation of Cln3 or Bck2 to activate SBF fully, and therefore, START is advanced sooner than in wildtype cells (**Fig 8K**).

The large size of *cln3Δ* can be explained by the absence of Cln3-activated SBF and MBF [86], and dependence on the less active Bck2-activated forms (SBFa5, and MBFa) (**Fig 8A**). Although SBF (unmodified; SBFa1) and Swi4 dimers (SBFa5) could contribute to the transcription factor pool, with Whi5 present, most of SBF is in a Whi5-bound inhibited complex. In agreement with this notion, additionally deleting Whi5 in *cln3Δ* cells makes them much smaller (closer to WT size), since SBF and Swi4 dimers (to a lesser extent) would now be present.

As discussed previously, Cln3 and Bck2 play mutually complementary roles in promoting the START transition (**Fig 8A–8B**). This is evident from the lethality of the double mutant *cln3Δ bck2Δ* [49]. In this mutant, both Cln3- and Bck2-activated forms of SBF and MBF are absent, while the inhibitor inhibits any available SBF (**Fig 8L**). The double mutant can

nevertheless be rescued by additionally deleting Whi5 (*cln3Δ bck2Δ whi5Δ*) [47,48], owing to the availability of unmodified SBF (SBFa1) that is free of inhibition and ready to transcribe (**Fig 8M**). These mutants are, however, larger than WT cells because they only depend on a less active form of SBF for transcription (due to lack of activators and resultant modifications; light green boxes in **Figs 2 and 3**).

Another set of mutants, *swi6Δ*, *cln3Δ*, and *swi6Δ cln3Δ*, can be understood with the help of the iconic representations shown in **Fig 8A, 8C,** and **8E**, and the simulations showing the redistribution of SBF and MBF forms. We note from the simulations that the mechanistic reasons explaining the viability of *swi6Δ* (2.25x) and *cln3Δ* (2.20x) are very different, even though they are both viable and large START mutants (**Fig 8A and 8C**). Although the *swi6Δ* mutant is viable due to the presence of Swi4 that can be activated by Bck2, the cells are extremely large (2.25x WT) due to the low activity of Swi4. In contrast, the viability of *cln3Δ* is due to MBF activated by Bck2 (**Fig 8A**). Furthermore, this explanation accounts for the similar size observed in *swi6Δ* and *swi6Δ cln3Δ* (~2.30x) cells [61] (**Fig 8C and 8E**). Both mutants depend on Bck2 for survival. Supporting evidence for our observation comes from the fact that both of these single mutants are lethal in a *bck2Δ* background [37,42] (**Fig 8F and 8L**). However, the observation that *swi6Δ CLN3-1* is also similar in size to *swi6Δ* and *swi6Δ cln3Δ*, indicates epistasis of Swi6 to Cln3 [61] (**Fig 8N**). Since Cln3 has secondary effects downstream of SBF (in START-BYCC), the sizes of these three mutants were not quite comparable to other experiments.

The results above suggest that deletion or over-expression of Cln3 would have no additional effect on *swi6Δ*, since the viability of *swi6Δ* only depends on the Swi4 dimer form (SBFa5) activated solely by Bck2 (**Fig 8E and 8N**). However, to pass the START transition, Cln1,2 (resulting from SBF transcription) and Cln3 are needed to i) inactivate Sic1, so that active Clb5,6 (transcribed by MBF genes) can accumulate to trigger DNA synthesis, and ii) inactivate Cdh1 to allow Clb2 to accumulate in preparation for mitosis. Thus, we reparameterized START-BYCC to take into account the lower effect of Cln3 on the cyclin antagonists (while ensuring that all other mutant phenotype emulations continue to hold). This also required changes in other parameters to ensure the viability of the double mutant *cln1Δ cln2Δ* (which depends on Cln3 and Bck2 to inactivate Sic1 for Clb5 to start DNA synthesis). Our current parameter set explains these three mutants' phenotypes, as well as the large size of *cln1Δ cln2Δ* (**S5 Table**). Here, again, we benefit from looking at the complexes present in each of these single, double, or triple mutants using the corresponding representative iconic tables and time-course simulations for different complexes.

Thus, we have parameterized our current model and adjusted the activity/contribution of different complexes such that the time-course simulations match the known experimental phenotypes as much as possible. Of the ~125+ mutant phenotypes that we simulate, we are able to reproduce most mutants to a high degree of accuracy (**S5 Table**). From among the remaining mutants, START-BYCC fails to explain only a few qualitatively, and one quantitatively. We will describe these contradictions in some detail below.

## Model inconsistencies

While the emulation of a predominant fraction of mutant phenotypes gives much credence to START-BYCC, the few contradictions point out gaps in our current understanding of the budding yeast cell cycle.

### swi6Δ GAL-WHI5

Whi5 only binds to intact SBF complexes and not to either Swi4 or Swi6 in isolation [48]. Accordingly, in START-BYCC, we do not allow Whi5 to bind to and inhibit the only available

active form in *swi6Δ*, the Swi4 complex activated by Bck2. Therefore, contrary to experimental findings [47], we expect *swi6Δ GAL-WHI5* cells to be similar in size to *swi6Δ* cells (**S9A Fig**). Such inhibition leads to the presence of free nuclear Whi5 molecules in *swi6Δ* cells, and their inhibition on Swi4 would reduce the propensity for swi6Δ to trigger START, making the single mutant too big to fit the experimental data. It is also likely that Whi5 has non-specific binding to START components, yet unexplored in this model.

### msn5Δ swi4Δ and msn5Δ swi6Δ

We expect the double mutants involving the export protein, Msn5, and SBF components, *msn5Δ swi4Δ* and *msn5Δ swi6Δ*, to be the same size as *swi4Δ* and *swi6Δ* cells, respectively (**S9B–S9E Fig**). This is because the only direct effect of Msn5 in START-BYCC is through the export of Whi5 and Swi6, which predominantly affects the localization of SBF, and, to a much lesser extent, MBF. In both these double mutants, there is no SBF to begin with. Therefore, additional deletion of Msn5 has no significant effect on either of these single mutants *swi4Δ* and *swi6Δ*. There is, however, a minor difference in size between *swi4Δ* and *msn5Δ swi4Δ*. The single deletion mutant depends only on MBF for its viability, and further deletion of Msn5 results in retaining phosphorylated Whi5 and Swi6 in the nucleus. Although dephosphorylation of these forms occurs in the nucleus, there is a substantial amount of phosphorylated Swi6 that cannot form MBF. The decrease in the amount of net MBF could explain the slightly larger size of *msn5Δ swi4Δ* as compared to the single deletion mutant of *swi4Δ* (**S9B–S9C Fig**). On the other hand, *swi6Δ* only depends on Swi4 for its viability. Therefore, the double mutant *msn5Δ swi6Δ* has the same size as *swi6Δ* cells (**S9D–S9E Fig**). This contradicts the experimental observation that these double mutants are inviable [85]. Clearly, START-BYCC cannot explain this scenario because we do not consider the other diverse effects of Msn5 on the cell, such as the transport of other phosphoproteins like Cdh1 [102]. Adding these interactions to the model in a subsequent update could help in explaining these mutants.

### cln1Δ cln2Δ cdh1Δ and cln1Δ cln2Δ cdh1Δ GAL-CLN2

Two other mutants that START-BYCC currently fails to explain are *cln1Δ cln2Δ cdh1Δ* and *cln1Δ cln2Δ cdh1Δ GAL-CLN2*. Contrary to experimental findings [93] and simulations from BYCC, the former triple mutant is inviable, while the latter quadruple mutant is viable yet very small in our simulations (**S9F–S9G Fig**). We suspect that this is due to abnormally high inhibition of CKI by Clb2 in the model, which is an artifact of re-parameterizing the model to fit several other START mutants. We found that lowering the efficiency of Clb2 inhibition on Sic1 indeed rescues the phenotypes, but results in other problems such as the viability of the lethal phenotype *swi4Δ swi6Δ*. The latter mutant does not have functional SBF or MBF, and therefore, no Cln1,2 or Clb5,6 and relies on Clb2 to keep CKIs' levels low. Therefore, the alteration of these parameters results in the cycling of *swi4Δ swi6Δ* cells. Similarly, lowering the efficiency of Clb2 on the other CKI, Cdc6, results in toggling the inviable mutant *sic1Δ cdh1Δ* to viable. This is because *sic1Δ cdh1Δ* depends on Clb2 to inhibit Cdc6. Lowering the efficiency would result in Cdc6 accumulation and reentry into G1. Therefore, even in this case, it is difficult to keep *cln1Δ cln2Δ cdh1Δ* viable and *sic1Δ cdh1Δ* inviable simultaneously. These scenarios let us posit that the balance and interplay between the mitotic cyclins and CKIs is more complicated than this that additional careful experiments might help resolve.

In *cln1Δ cln2Δ cdh1Δ* mutant, cells are very large at the time of mitotic exit due to *cln1Δ cln2Δ*, and high Clb2 (due to *cdh1Δ* and high mass). High Clb2 inhibits the accumulation of CKI needed for *cdh1Δ* cells to exit from mitosis. Therefore, the model was modified to include additional experimental details on Cdh1's effect on Cdc20 degradation. Effectively, this results

in a reduction of Clb2 levels in the absence of Cdh1. Despite several modifications made in the parameters and wiring of the model, the current model still does not include the effect of Cdh1 on Polo kinase (Cdc5). If we consider these effects, then in the *cdh1Δ* mutant, Polo would be stabilized, causing Cdc14 release and, in turn, resulting in CKI synthesis. Thus, with higher CKI and a lower amount of Clb2, the mutant *cln1Δ cln2Δ cdh1Δ* might be able to exit mitosis in line with observed experiments.

Currently, in START-BYCC, *cln1Δ cln2Δ cdh1Δ GAL-CLN2* is viable but extremely small (**S9G Fig**). *GAL-CLN2* inhibits CKI, which increases at the mitotic exit. If the efficiency of CLN2 inhibition on CKI is lowered, then the mutant *cln3Δ* grows too big and dies. This is because MBF active in *cln3Δ* cells makes a small amount of Cln2, and the viability of the mutant relies heavily on Cln2 to inhibit Sic1. This inhibition in *cln3Δ* would result in Clb5 accumulation and DNA synthesis. Therefore, it is quite difficult to achieve simultaneous viability of both *cln1Δ cln2Δ cdh1Δ GAL-CLN2* and cln3Δ. Here again, incorporating the effect of Cdh1 on Polo and Cdc14 in START-BYCC might make the mutant viable due to higher levels of CKI (the viability of *cln1Δ cln2Δ cdh1Δ GAL-CLN2 GAL-SIC1* (**S9H Fig**) supports our hypothesis).

We predict that a future version of START-BYCC with a more detailed version of mitotic exit would successfully explain these mutants, as well as several others.

The advantages of our detailed mathematical model extend beyond understanding the molecular mechanisms underlying known mutant phenotypes. It also summarizes, organizes, and reconciles current knowledge about cell cycle regulation, from which we can make testable predictions (see tested predictions here [101]). In this section, we enumerate some of the predictions from START-BYCC (**S10 Fig**) and provide the entire list as part of **S5 Table** (highlighted in blue). More of these predictions can be explored in our online simulator: http://www.sbmlsimulator.com/simulator/by-start.

### bck2Δ mbp1Δ

Consider the following cases: deletion of Bck2 and Cln3 in *mbp1Δ* background. Based on the sizes of *cln3Δ* and *bck2Δ* (*cln3Δ >> bck2Δ*), we infer that Cln3 is a more efficient activator of SBF and MBF than Bck2. Similarly, based on the sizes of *swi4Δ* and *mbp1Δ*, SBF is more active than MBF. We, therefore, expect the mutant **bck2Δ mbp1Δ** to be viable since the major activator of SBF, Cln3, is still present (**S10A Fig**). In the simulation, we observe that the Cln-activated SBF forms are present (SBFa2>SBFa3, SBFa4, SBFa1) and that they contribute to the viability of the cell in the absence of MBFa and Bck2, in agreement with the validated phenotype in Adames et. al., 2015 [101].

### bck2Δ mbp1Δ GAL-WHI5

Even though Whi5 is an inhibitor of SBF, *GAL-WHI5* need not always be highly detrimental to the cell. This is because we expect the G1 cyclin (Cln3) and the downstream cyclins that engage in a positive feedback loop (Cln1,2 and Clb5,6) to efficiently phosphorylate the additional Whi5-bound SBF added to the system. We therefore expect that in the mutant **bck2Δ mbp1Δ GAL-WHI5**, the Cln-activated forms (SBFa2>>SBFa3, SBFa4) would contribute to cell viability (**S10B Fig**), but the experimental phenotype shows that this mutant is G1 arrested [101]. Since we assign full activity to Whi5-bound SBF that is phosphorylated on Swi6 by Cln kinases, SBFa3 and SBFa4 will contribute to overall SBF activity, and the triple mutant will be viable. Similar to this triple mutant, the double mutants **mbp1Δ GAL-WHI5** and **mbp1Δ GAL-WHI5-12A** are also viable despite excessive Whi5 in the system because we expect that they are converted to active Whi5-bound SBF forms (SBFa2, SBFa3, SBFa4 in the former case

and SBFa3 in the latter non-phosphorylable case) (**S10H–S10I Fig**). These forms are present at sufficiently high levels to ensure viability. Very high doses of Whi5 might saturate the kinases in the system, resulting in cell inviability. For other moderate increases in Whi5 or Whi5-12A levels, we expect the aforementioned response in the background of *mbp1Δ* cells.

### cln3Δ mbp1Δ, cln3Δ mbp1Δ swi6Δ, and cln3Δ mbp1Δ whi5Δ

In START-BYCC, we would expect the double mutant ***cln3Δ mbp1Δ*** to be viable and large (with a delayed START for cycle 1). This is because, with no MBF, only SBF is available as the activator of Cln1,2 and Clb5,6, which delays the cycle and results in a larger cell size. The triple deletion mutant cln3Δ mbp1Δ swi6Δ is predicted to be viable and large with neither SBF nor MBF. Only swi4 dimers exist (only SBFa5 is present). Since swi4 dimers can only be activated by Bck2, this delays the cell cycle, enhancing cell growth (**S10D Fig**; almost the same size as cln3Δ mbp1Δ). We predict another triple deletion mutant, cln3Δ mbp1Δ whi5Δ, to reduce the cell size (from cln3Δ mbp1Δ). As seen in **S10E Fig**, the size of cells is smaller because whi5 is not present, so there is no inactivation, and SBF activations happen much earlier. All of our simulations here agree with the experimentally validated phenotypes [101].

### Rescue of cln3Δ swi4Δ by whi5Δ, GAL-BCK2, and whi5Δ sic1Δ

The double mutant ***cln3Δ swi4Δ*** is inviable because the only active form present is MBF activated by Clns or Bck2 [103]. Our simulations show that an additional deletion of Whi5 (***cln3Δ swi4Δ whi5Δ***) can rescue the phenotype, as MBF is relieved in the absence of Whi5 (**S10F Fig**), in line with experimental validation [101]. The mutant ***cln3Δ swi4Δ whi5Δ*** is predicted to be further rescued by the additional deletion of Sic1 due to early relieving of inhibition on active MBF (this quadruple mutant with *sic1Δ* is smaller than the parent triple deletion). Similarly, overexpression of Bck2 also rescues the double mutant that lacks SBF, because there is sufficient Bck2-activated MBF (***cln3Δ swi4Δ GAL-BCK2***, MBFa forms in **S10G Fig**) to drive forward the cell cycle.

### Rescue of swi4Δ swi6Δ by GAL-CLB5 and GAL-CLN2 (not by GAL-CLN3)

Both the double mutants *swi4Δ swi6Δ and swi4Δ mbp1Δ* are inviable in our simulations due to the absence of SBF/MBF, in strong agreement with experimental observations [21,23] (**Fig 8**). The logical prediction from the model follows that an additional deletion of Whi5 would not rescue the double deletion of Swi4 and Swi6; the triple deletion *swi4Δ swi6Δ whi5Δ* is inviable (**Table 1**). These cells do not have any SBF or MBF, and hence Whi5 does not have any stoichiometric partner to bind with and inhibit. However, ***swi4Δ swi6Δ*** can be rescued by ***GAL-CLB5*** and ***GAL-CLN2*** (but not by ***GAL-CLN3***), since CKIs are still high enough to prevent triggering DNA synthesis (**S5 Table**). Similarly, *GAL-CLB5* and *GAL-CLN2* can also rescue *bck2Δ swi6Δ* (but not ***GAL-CLN3***) (**S5 Table**).

### Rescue of swi4Δ swi6Δ and bck2Δ swi6Δ by SWI6-SA4

Similar to rescue by multi-copy Swi6 [49], we also expect the lethal mutants ***swi4Δ swi6Δ* and *bck2Δ swi6Δ* to be rescued by *SWI6-SA4***, since MBF would be available again in the former case, and both SBF (although non-phosphorylable at Swi6) and MBF available in the latter case, to rescue the lethal phenotype (**S5 Table**).

## Conclusion

The cell cycle circuit is the central machinery of the cell, driving its controlled growth and division. The START transition in the budding yeast cell cycle serves as an important checkpoint

to ensure the absence of inhibitory signals and the growth of the cell to a critical size before commitment to cell division. In this study, we developed a detailed mathematical model of the START transition (START-BYCC), incorporating molecular details of this process, including signaling/regulatory interactions, phosphorylation states, and subcellular localization of key START proteins. The model recapitulates the experimentally observed phenotypes of over 120 mutants pertaining to START and other important cell cycle phases (several representative mutants can be accessed here: http://www.sbmlsimulator.com/simulator/by-start). START--BYCC also captures the subcellular localization and translocation of the transcription factor, SBF, and provides a mechanism for size control.

Most importantly, the analysis we present here establishes a foundation for novel hypotheses and precise predictions on the mechanisms of the START transition that can be verified experimentally (including some successful validations highlighted recently [101]). Secondly, the nutritional conditions of a cell greatly influence its size control mechanism [19], including by several recently discovered molecular players mediating this effect [19,58,70,71]. These interactions, and any newer findings, can be dovetailed into START-BYCC to generalize the existing mechanism for size control. Future work will entail appending detailed current models of other phases and events of the budding yeast cell cycle, including DNA damage response, spindle assembly checkpoint, and mitotic exit. START-BYCC provides an opportune setting to incorporate these modules (the model is available here for any future extensions and expansions: github.com/jravilab/start-bycc). Finally, this detailed model for budding yeast offers a path towards dealing with the enormous complexity of the cell cycle mechanism in high eukaryotes, including humans allowing us to uncover the fundamental principles and regulatory bottlenecks that are essential to delineate the underlying system dynamics. The evolutionary conservation of the molecular circuitry underlying the START transition in the yeast cell cycle and R-point in mammalian counterparts supports this view, and our detailed model would help these parallel eukaryotic studies in bridging our gaps in understanding these complex systems. Modeling the R-point not only offers an understanding of the workings of a healthy cell cycle, but also tenders insights into its characteristic deregulation in diseases like cancer.

## Materials and methods

### Current START-BYCC model

A large number of experimental findings were reconciled to establish a detailed wiring diagram for the START transition (**Figs 2 and 3**) in conjunction with the BYCC model for the budding yeast cell cycle. Several new aspects of the regulation of START have been revealed since 2004, including the role of the protein Whi5, which is a stoichiometric inhibitor of the transcription factor SBF. The START module of the model has been described in detail under the 'Results' section (**Figs 2 and 3**). The remaining cell cycle model (from START to mitotic exit) has been used almost as-is from the BYCC model (except for changes in equations of START proteins like Cln3, Bck2, Cln2, and Clb5). The wiring diagram was converted to chemical reactions (mass-action) and algebraic relations, which were then translated to ordinary differential equations and algebraic equations. The codes, wiring diagrams, equations, and parameters will be available via GitHub (https://github.com/jravilab/start-bycc) and our easy-to-use online simulator (http://www.sbmlsimulator.com/simulator/by-start).

### Modifications from the BYCC model

We started with the well-established budding yeast cell cycle (BYCC) mathematical model from Chen et al., 2004 [17]. The major modifications are associated with the START transition of the

cell cycle. The simplified Goldbeter-Koshland switch, previously used to describe the activation of SBF and MBF in the BYCC model, has been replaced with the substantially detailed model presented in **Figs 2 and 3**. This new model delineates several important phosphorylation and transportation events occurring at START. Cellular compartments—nucleus and cytoplasm—are explicitly modeled to track localization. They are assumed to have a constant volume ratio of 1:4 throughout the cell cycle [95,104]. The species that move across compartments are scaled with the corresponding volumes to make the concentrations in the equations consistent.

Additionally, in the current version of the model, we have removed mass dependence from cyclins such as Cln1,2 represented as a single variable in the model, Cln2, since they are not concentrated in the nucleus and are distributed roughly throughout the cell. We have retained mass dependence only on Cln3 (through Ydj1 as the size sensor and on Cln3 level in the nucleus), Bck2, Clb5, and Clb2 in order to explain the complicated contributions of these species to size control and/or their nuclear localization. We also use a Hill function to model Cln activation at START (equation of Vpcln in **S2 Text**) due to the existence of cooperativity [90], and to avoid the complexity and additional intermediates that would stem from considering multi-site phosphorylation of Whi5, Swi6, and Swi4. We tested a simpler version of the START module modeled with multi-site phosphorylation and confirmed that it showed a behavior similar to the Hill function-based model. Thus, we concluded that the Hill function is a reasonable approximation and phenomenological abstraction of multi-site phosphorylation in our case, and used it in START-BYCC.

## Equations and parameters

The equations and parameters used for our simulations are listed in **S2 Text**, and the files have been provided in the following formats: ODE, SBML, and PET, including the parameter set used. We have also listed all changes made to the initial conditions and parameters in simulating specific mutants (**S2 Table**), along with key assumptions (**S3 Table**). The codes, equations, and parameters will be available via GitHub (github.com/jravilab/start-bycc) and our online simulator (http://www.sbmlsimulator.com/simulator/by-start). The equations can also be viewed here through the rendered markdown PDF in our repository.

## Simulations

We used JigCell [105] to build START-BYCC by defining biochemical reactions (or interactions), algebraic rules, conservation relations, discrete events at the end of the cell cycle, and multiple compartments. All reactions, equations, and conservation relations were verified manually. Parameter Estimation Toolkit (PET) was used to run all our simulations (using LSODAR) [106] for wildtype and mutants by defining the specified conditions in **S4 Table**. The software offers the option of simultaneously running simulations for different conditions (termed 'simulation runs') and for several sets of rate constants (termed 'basal sets'). XPPAUT [107] (sites.pitt.edu/~phase/bard/bardware/xpp/xpp.html) was also used to check our simulations numerically (for wildtype cells).

## Supporting information

**S1 Fig. Complex formation & promoter binding.** Initial flow of events in G1 starting from (1) monomers. The top panel is the core model for SBF activation and inactivation as appeared in Fig 2A. The red box containing reactions involved in (2) the complex formation and (3) promoter binding is expanded and shown in the lower panel.
(PDF)

**S2 Fig. SBF regulation in wildtype cells.** (i) SBF activation and Whi5 export. In the figure, the cells start with monomers, which proceed to form complexes and bind to the promoter. In WT cells, both Swi6 (P-form) and Whi5 get phosphorylated by Clns (Cln3, Cln1,2 and Clb5,6) and activated by Bck2. We assume that this doubly phosphorylated form is unstable and dissociates to yield active SBF (with Swi6 phosphorylated) and phosphorylated Whi5 that is free to move to the cytoplasm with the help of export protein, Msn5. (ii) SBF inactivation and export. SBF gets inactivated by two sets of Clb phosphorylations (black-filled circles): Q-form on S160 site of Swi6 by Clb5,6 and Clb1,2, and on Swi4 by Clb1,2, leading to dissociation of SBF from promoter. Msn5 recognizes the Q-form of Swi6 phosphorylation (that is in complex with Swi4) for export to the cytoplasm. Complex dissociates in the cytoplasm soon after export. Phosphate groups are indicated as white-filled circles (activatory phosphorylations) or with black-filled circles (inhibitory phosphorylations).
(PDF)

**S3 Fig. SBF inactivation by Clbs.** We assume that either Clb phosphorylation on Swi6 (S160) (Q-form), or Cln phosphorylation on Whi5 is necessary for recognition by transport protein, Msn5, and subsequent export to the cytoplasm. Additionally, we assume that in promoter-bound complexes, Clb phosphorylation of Swi4 is necessary and sufficient for the complexes to dissociate from the promoter necessary to turn off gene transcription and to facilitate SBF export. Similar to SBF inactivation in S2, SBF Fig complexes dissociate in the cytoplasm soon after export.
(PDF)

**S4 Fig. Cartoon of Cln3 regulation by Ydj1/Ssa1.** The nuclear form of Cln3 is the active form. Nuclear import is controlled by Ydj1 in response to cell size, whereas sequestration of Cln3 in the ER (inhibiting Cln3) is done by Ssa1, probably, in response to late mitotic factors. (See main text for details) We assume that Bck2 is regulated in a similar fashion.
(PDF)

**S5 Fig. Duration of daughter cycle times as a function of mass doubling time.** The duration plots of daughter cycle times (green-filled squares) and G1 (red-filled triangles) phase versus mass doubling time correspond to the experiments by Lord and Wheals (1980). The budded period (blue-filled circles) was, however, close to 50 and almost constant over the wide range, differing quantitatively from the corresponding curve in the experiments.
(PDF)

**S6 Fig. Localization of different monomers.** This figure describes the timing of re-import and hence overall temporal localization of the different monomers (Swi4, Swi6, Whi5). Export and reimport of Whi5P, although not shown explicitly, follows the same steps as phosphorylated Whi5 in the cartoon. Step (1): SBF complexes that have been phosphorylated on Whi5 or the S160 site of Swi6 are transported to the cytoplasm by Msn5 and dissociate immediately. Phosphorylated Whi5 monomers are also exported by Msn5. Step (2): Unphosphorylated monomers move back to the nucleus (regardless of the phase of the cell cycle). Step (3): Swi4 and the P-form of Swi6 (all phosphorylation sites except S160) get dephosphorylated by PP2A and move to the nucleus. Step (4): the phosphatase Cdc14 that accumulates at mitotic exit dephosphorylates Whi5 and Swi6 Q-form at residue S160, following which Whi5 and Swi6 get reimported to the nucleus resetting the localization state for the G1-phase of the next cell cycle.
(PDF)

**S7 Fig. Importance of the transport protein, Msn5.** These simulations are plotted for msn5Δ cells (MSN5 = 0 in our model). The cells are slightly larger than WT cells due to the absence of

active SBF, and lesser amount of active MBF (MBFa in the plot above).
(PDF)

**S8 Fig. Non-phosphorylable mutants.** Active complexes in (A) *WHI5-12A* (This mutant will have size similar to that of WT due to the Swi6 P-forms; Fig 7A), (B) *SWI6-SA4* (WT size due to phosphorylation of Whi5; Fig 7B), (C) *WHI5-12A SWI6-SA4* (Viable, yet large, due to inactive SBF-Whi5 complex and support from Bck2 activation and MBF; Fig 7C).
(PDF)

**S9 Fig. Model contradictions.** The figure shows simulations of the following mutants: (A) *swi6Δ GAL-WHI5* (only Swi4dimers present (SBFa5); cells are viable yet large), (B) *swi4Δ* (only MBF present; cells are viable yet large), *(C) msn5Δ swi4Δ* (only MBF present; cells are viable and large), (D) *swi6Δ* (only Swi4dimers (SBFa5) present; cells are viable yet large), *(E) msn5Δ swi6Δ* (only Swi4dimers (SBFa5) present; cells are viable yet large), (F) *cln1Δ cln2Δ cdh1Δ* (cells are inviable), (G) *cln1Δ cln2Δ cdh1Δ GAL-CLN2* (cells are viable, very small). A, C, E, F and G are in contradiction with experimental findings, while (H) *cln1Δ cln2Δ cdh1Δ GAL-CLN2 GAL-SIC (cells are viable)* is complementary with our hypothesis about the effect of Cdh1 on Polo and Cdc14 on cell viability due to higher CKI levels.
(PDF)

**S10 Fig. Few model predictions and validations.** (A) *bck2Δ mbp1Δ* (Cln-activated forms of SBF are present in varying fractions (SBFa2>SBFa3, SBFa4> SBFa1); cells are viable), (B) *bck2Δ mbp1Δ GAL-WHI5* (Cln-activated forms of SBF are present (SBFa2, SBFa3, SBFa4) even if they are lesser than in A; cells are still viable), (C) *cln3Δ mbp1Δ* (SBFa2 > SBFa3 >> SBFa4 > SBFa5; no MBF is present; cells are viable), (D) *cln3Δ mbp1Δ swi6Δ* (Swi4dimers (SBFa5) are present sufficiently enough to rescue cells; cells are viable), (E) *cln3Δ mbp1Δ whi5Δ* (Inhibition on SBF is relieved); cells are rescued), (F) *cln3Δ swi4Δ whi5Δ* (More Bck2-activated MBF is present (MBFa), there's no SBF; cells are viable), (G) *cln3Δ swi4Δ GAL-BCK2* (There's a sufficient amount of Bck2-activated MBF (MBFa); cells are rescued), (H) *mbp1Δ GAL-WHI5* cells are viable since Whi5 is predicted to get activated and is present in the forms of SBFa2, SBFa3, and SBFa4 (I) *mbp1Δ GAL-WHI5-12A* cells are viable because Whi5 get converted to the activated form SBFa3.
(PDF)

**S1 Table. Functions of different cyclins in budding yeast cell cycle.**
(PDF)

**S2 Table. Description, abundance, regulation, and localization of START components in the model.**
(PDF)

**S3 Table. List of Key Assumptions.**
(PDF)

**S4 Table. Modifications in parameter and initial conditions corresponding to mutants. (Mutants exclusive to the current model are emphasized in bold).**
(PDF)

**S5 Table. START mutants.**
(PDF)

**S1 Text. List of abbreviations.**
(PDF)

**S2 Text. Equations, parameters, and initial conditions.**
(PDF)

## Acknowledgments

First and foremost, we would like to thank John Tyson (now emeritus) for invaluable advice, discussion, and support during the early stages of the project. We would like to thank Kartik Subramanian, Debashis Barik, Sandip Kar, Tongli Zhang, Teeraphan Laomettachit, Vandana Sreedharan, and Arjun Krishnan for providing the authors with several iterations of constructive feedback. We are also extremely grateful to Krishnan Raghunathan, Arjun Krishnan, and Emily Meyer for their detailed comments on the manuscript. We especially appreciate Evan Brenner's timely comments and detailed feedback on the manuscript. We have benefited from several conversations and diverse mutant phenotypes and challenges brought to us by researchers in the budding yeast cell cycle field that helped us fine-tune the model, including Jan Skotheim, Robert De Bruin, Stefano Di Talia, and Fred Cross. We are also grateful to Jean Peccoud and their team for the first set of experimental validations [101].

## Author Contributions

**Conceptualization:** Janani Ravi.

**Data curation:** Janani Ravi, Kewalin Samart.

**Formal analysis:** Janani Ravi, Kewalin Samart.

**Funding acquisition:** Janani Ravi.

**Investigation:** Janani Ravi, Kewalin Samart.

**Methodology:** Janani Ravi.

**Project administration:** Janani Ravi.

**Resources:** Janani Ravi.

**Software:** Janani Ravi, Jason Zwolak.

**Supervision:** Janani Ravi.

**Validation:** Janani Ravi.

**Visualization:** Janani Ravi, Kewalin Samart, Jason Zwolak.

**Writing – original draft:** Janani Ravi.

**Writing – review & editing:** Janani Ravi, Kewalin Samart.

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
