## [Decision Letter · Decision Letter 0]

6 Dec 2023

Dear Dr Ravi,

Thank you very much for submitting your manuscript "Modeling the START transition in the budding yeast cell cycle" for consideration at PLOS Computational Biology.

As with all papers reviewed by the journal, your manuscript was reviewed by members of the editorial board and by several independent reviewers. In light of the reviews (below this email), we would like to invite the resubmission of a significantly-revised version that takes into account the reviewers' comments.

Please address the points raised by the referees about controversial assumptions of the model and describe properly the choices of model assumptions with proper references.

We cannot make any decision about publication until we have seen the revised manuscript and your response to the reviewers' comments. Your revised manuscript is also likely to be sent to reviewers for further evaluation.

Sincerely,

Attila Csikász-Nagy

Academic Editor

PLOS Computational Biology

Pedro Mendes

Section Editor

PLOS Computational Biology

Reviewer's Responses to Questions

**Comments to the Authors:**

Reviewer #1: The article by Ravi et al describes a new version of the cell cycle model that was initially developed many years ago for budding yeast cells and pioneered the powerful application of systems biology to a complex biological process. This model has been subject to periodic revisions to accommodate the new knowledge attained in this active field of research and, in this occasion, the authors focus their attention to the molecular interactions that control START in late G1. As in prior versions, the model is very well developed and tested, and the needed assumptions are reasonable in general terms. There are, nevertheless, important experimental data that should be adopted by the model to attain a more realistic description of the triggering factors of START. In addition, some issues should be clarified or solved in the text.

1. “In early G1, only Cln3 is available, and its level increases with cell size (ref. 2)”. This is still a matter of debate and refs for experimental evidences in favor and against should be provided. Please also check ref 2.

2. Dilution of Whi5 in G1 has been proposed as a relevant sizer at START (10.1038/nature14908). This mechanism should be underlined in the text and, preferably, added to the model. Also, in table S2 the authors state that “Whi5 is known to be transcriptionally regulated showing a 3-fold variation, peaking in late G1 phase (ref. 4)”. By contrast, WHI5 is more expressed in G2/M phases (10.1038/nature14908).

3. “Ydj1 is believed to act as the cell size sensor”. However, Ssa1 and Ydj1 chaperones have been involved in cell size control as growth rate sensors (ref. 79), not as reporters of size per se. In addition, in table S2 the authors state that “Ydj1, which is proposed to be the sensor of cell size (ref. 18)”. Please check the reference and clarify.

4. Degradation of Cln3 by SBF-Cdc4/Mad3 has been shown to cause a progressive accumulation of Cln3 in the nucleus during G1 (10.1126/sciadv.abm4086). As this mechanism connects START to the preceding cell cycle, it should be included in the model as a sizer mechanism. In addition, the authors could test whether activation of the spindle checkpoint predicts a larger cell size in the subsequent START event as observed.

5. Although not strictly required in the model, RNAPol2 phosphorylation by Cdc28/Cln3 should be considered in the text (10.1126/science.aba5186).

Reviewer #2: In the manuscript entitled ‘Modeling the START transition in the budding yeast cell cycle’ by Ravi et al, the authors proposed a detailed dynamical model of START transition in the Saccharomyces cerevisiae cell cycle. Their model of START transition is integrated into the well-established previous model of budding yeast cell cycle (Chen et al, Mol. Biol. Cell 15, 3841(2004)). The model takes into account of key details of activation, inactivation and localization of several regulatory factors Swi4, Swi6, SBF, Whi5, MBF and Bck2 to explain key mutant phenotypes associated with the START transition in the budding yeast cell cycle. The model indeed captures phenotypic behaviour of many mutants relevant to the START transition.

The manuscript has great merit for publication the PloS Computational Biology. I have a few queries and suggestions for the authors to consider in their revision.

1. For a mathematical model paper, it is extremely important to have the model equations listed in a table either in the main or supplementary text. These equations should be listed in the readable format in a table instead of .ode or .pet format as supplementary file. Supplementary files (.ode or .pet) are required for reproducibility.

2. Single cell quantitative experiments on budding yeast cell cycle have shown that the average cycle time of daughter and mother cells, respectively, are 112 and 87 min in glucose medium for the WT cells (Di Talia et al, Nature 448, 947 (2007)). In addition, the average duration of the unbudded phase for them were found to be 37 and 16 min, respectively. Although the model nearly recaptures the cell cycle duration (107 min) of the daughter cell however it reports a significantly longer duration of G1 phase (62 min) and consequently the SG2M phase (or budded phase) becomes quite short. Therefore the authors need to justify the gap between the model and the available experimental data. Furthermore the details of mother-daughter distinction at the cell division need to be mentioned.

3. Previous experimental (Schmoller et al, Nature 526, 268 (2015)) and modelling (Heldt at al, PLoS Comput Biol 14(10): e1006548 (2018)) studies have uncovered a new perspective of size control in budding yeast cell cycle. According to these studies dilution of cell cycle inhibitor (Whi5) and titration of cell cycle activator (Cln3) become key driving factor for the size control in budding yeast. It will be worth discussing the feasibility and connection of dilution-titration mechanism in this model.

4. The Figure S5 gives a false impression of mismatch between the MDT and the cell cycle time, as it plots the cell cycle time of the daughter cell with the MDT of the population consisting of both mother and daughter cell.

5. For better readability, the authors many think about alternative way of reporting the results of Figure 8 as the plots spread across multiple pages.

**Have the authors made all data and (if applicable) computational code underlying the findings in their manuscript fully available?**

Reviewer #1: Yes

Reviewer #2: Yes

PLOS authors have the option to publish the peer review history of their article (what does this mean?). If published, this will include your full peer review and any attached files.

Reviewer #1: No

Reviewer #2: No
---

## [Decision Letter · Decision Letter 1]

2 Apr 2024

Dear Dr Ravi,

We are pleased to inform you that your manuscript 'Modeling the START transition in the budding yeast cell cycle' has been provisionally accepted for publication in PLOS Computational Biology.

Best regards,

Attila Csikász-Nagy

Academic Editor

PLOS Computational Biology

Pedro Mendes

Section Editor

PLOS Computational Biology

Reviewer's Responses to Questions

**Comments to the Authors:**

Reviewer #1: The authors have addressed the most important points raised in my initial revision, and I truly acknowledge their effort to make their review as comprehensive as possible, particularly given the highly complex functional interactome controlling the cell cycle.

Reviewer #2: The authors have provided satisfactory justifications to my queries and comments. I recommend acceptance of the manuscript for publication.

**Have the authors made all data and (if applicable) computational code underlying the findings in their manuscript fully available?**

Reviewer #1: Yes

Reviewer #2: Yes

PLOS authors have the option to publish the peer review history of their article (what does this mean?). If published, this will include your full peer review and any attached files.

Reviewer #1: No

Reviewer #2: No

---

## [Editor Report · Acceptance letter]

29 Jul 2024

PCOMPBIOL-D-23-01835R1 

Modeling the START transition in the budding yeast cell cycle

Dear Dr Ravi,

I am pleased to inform you that your manuscript has been formally accepted for publication in PLOS Computational Biology. Your manuscript is now with our production department and you will be notified of the publication date in due course.

With kind regards,

Anita Estes
